# HalDec-Bench: Benchmarking Hallucination Detector in Image Captioning

## Abstract

Recent progress in large vision-language models (VLMs) has been driven by advances in image-text alignment, i.e., learning the relationship between image and text. Hallucination detection in captions, **HalDec**, is a task to assess VLM's image-text alignment ability, and aims to identify errors in VLM-generated captions that misrepresent image content. Detecting these errors is crucial not only for evaluating alignment ability but also for curating high-quality image-caption pairs used to train VLMs. While VLMs have been explored as hallucination detectors, their generalizability across different captioning models, image domains, and hallucination types remains unclear due to a lack of a benchmark. In this work, we present HalDec-Bench, the first benchmark for principled and interpretable evaluation of HalDec models. It covers diverse VLMs used as captioning models, image domains, and provides high-quality hallucination-existence annotations enriched with hallucination-type labels. HalDec-Bench thus serves as a comprehensive testbed to advance HalDec and probe the image-text alignment ability of VLMs. Our analysis shows that HalDec-Bench offers tasks of varying difficulty, making it well-suited as a HalDec benchmark. Evaluating diverse VLMs reveals key limitations: (i) CLIP-like models are nearly blind to hallucinations in recent VLMs, (ii) detectors tend to over-score early sentences, and (iii) they display strong self-preference—favoring their own captions—which undermines detection performance. We will release our evaluation code and dataset upon acceptance.

## 1 Introduction

We have seen remarkable progress in large vision-language models (VLMs) (Wang et al., 2024a; Liu et al., 2023; 2024; Chen et al., 2023; Li et al., 2023a) and text-to-image generative models (Betker et al., 2023). A key to this progress lies in understanding image content in the form of text, i.e., learning image-text alignment. Once this mapping between images and text is effectively learned, large language models (LLMs) can be leveraged for various visual reasoning tasks (Li et al., 2023a).

Hallucination detection in captions, called **HalDec** hereafter, is a task that assesses VLM's image-text alignment capability. It aims to identify errors in captions that misrepresent image content, such as misstated object counts, incorrect attributes or relationships, or the introduction of entities absent from the image (Biten et al., 2022; Li et al., 2023b; Rohrbach et al., 2018). Beyond evaluating the alignment ability of VLMs, HalDec enables filtering out unaligned image-caption pairs (Li et al., 2022) from the training data. In practice, VLM training often relies on captions synthesized by a *Captioner*[1] to supplement the limited availability of human-annotated data. However, these synthetic captions frequently suffer from hallucinations. Curating high-quality image-caption pairs with strong detectors, therefore, plays a crucial role in building performant VLMs (Chen et al., 2024a). Indeed, models such as CLIP (Radford et al., 2021) and BLIP (Li et al., 2022) have already been widely used to curate large-scale training datasets for VLMs (Betker et al., 2023; Li et al., 2023a).

Considering the scalability of detectors, we expect the detector to be universally applicable across diverse image-caption pairs. Thus, as shown in Fig. 1, evaluating HalDec requires testing models to detect hallucinations across different Captioners, image domains, and hallucination types, since

---

[1]To avoid confusion, we use the term *Captioner* to denote a VLM used for caption generation, and use *Detector* to denote a VLM used for hallucination detection model.

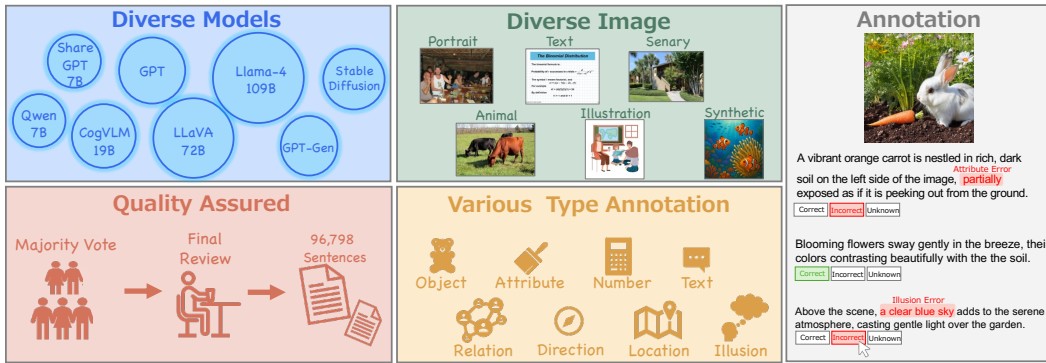

Figure 1: We introduce a novel benchmark, **HalDec-Bench**, for evaluating hallucination detectors on image captions generated by VLMs. Beyond measuring the effectiveness of hallucination detection in image captions, the benchmark also probes VLMs' ability to capture fine-grained image-text alignment. It spans a diverse set of captioning VLMs (top left) and image domains (top center) , and provides high-quality annotations (bottom left) enriched with hallucination type labels (bottom center).

each factor can introduce distinct language styles and error patterns. Yet, the universality of large VLMs as a hallucination detector remains unclear, even though recent studies have fine-tuned them for hallucination detection (Gunjal et al., 2024; Wada et al., 2025). Similarly, CLIP has been developed to learn the relationship between an image and a detailed sentence (Patel et al., 2024; Yuksekgonul et al., 2023), but its effectiveness is not clear for captions synthesized by advanced Captioners. The challenge of making the comprehensive HalDec benchmark is to build a dataset suited for such evaluation, requiring a significant cost of human annotation, where annotators must carefully check the image-sentence alignment. In fact, existing HalDec datasets suffer from limited model coverage and insufficient scale. MHalDetect (Gunjal et al., 2024) provides annotations for only a single VLM, and MHaluBench (Chen et al., 2024b) covers only a small set of models and samples. Also, despite the development of benchmarks for multimodal reasoning (Yue et al., 2024; 2025; Liu et al., 2023; Lu et al., 2023), a benchmark to evaluate VLMs' fundamental image-caption alignment is limited to hallucinated sentences generated by a human-designed pipeline (Yuksekgonul et al., 2023; Hsieh et al., 2023).

In this paper, we introduce HalDec-Bench, a benchmark designed to evaluate hallucination detectors for image captions in a principled and interpretable manner. As illustrated in Fig. 1, it covers a diverse set of captioning VLMs and image domains, and provides high-quality annotations enriched with hallucination-type labels and segment-level annotations. Beyond serving as a tool for analyzing detectors, HalDec-Bench also functions as a testbed for probing VLMs' fundamental ability to capture image-caption alignment. In our experiments, we focus on sentence-level hallucination detection and assess a variety of VLMs as detectors. The results demonstrate that HalDec-Bench offers tasks with diverse levels of difficulty, making it well-suited as a HalDec benchmark. Our extensive analysis further yields several key insights. First, detectors tend to recognize sentences at the beginning of a response as *correct*, regardless of their correctness. Second, they exhibit self-preference, i.e., consider their own output captions as *correct*, which degrades performance as detectors. This observation is consistent with prior findings (Panickssery et al., 2024). Third, we show that diverse ensembling strategies can effectively improve HalDec performance.

## 2 RELATED WORK

**Benchmarks for VLMs.** Many benchmarks evaluate the broad reasoning ability of VLMs (Yue et al., 2024; 2025; Guan et al., 2024; Fu et al., 2023; Li et al., 2024b; Tong et al., 2024a) or expert knowledge with visual inputs (Lu et al., 2023). Some quantify VLMs for questions that are designed to hallucinate VLMs (Guan et al., 2024; Wang et al., 2023a). While prior datasets evaluate whether questions can mislead VLMs, we instead assess their capability to detect hallucinations in image captions, thereby emphasizing understanding of image-text alignment. Tong et al. (2024b) test fine-grained visual comprehension using CLIP-blind image pairs and related questions. HalDec-Bench instead utilizes captions from advanced VLMs, whose errors are difficult to detect.

**Hallucination detection and mitigation in image captioning.** Hallucination detection in image captioning has been widely studied (Rohrbach et al., 2018). CHAIR (Rohrbach et al., 2018) was the first metric to evaluate image-caption alignment at the object level using an object detector. However, its effectiveness is constrained by the detector's coverage and accuracy; thus, it fails in capturing the diverse hallucination types and captioning styles. Also, many works attempt to mitigate the hallucinations in image captions (Zhang et al., 2024; Leng et al., 2024; Farquhar et al., 2024; Zhou et al., 2024; Zhuang et al., 2025; Favero et al., 2024; Woo et al., 2025; Suo et al., 2025), especially, mitigating hallucinations in long captions is important as they are prone to contain more hallucinations (Zhou et al., 2024; Hirota et al., 2025), which we confirm in Sec. 4.2. Refining a captioning model based on image-caption alignment score, computed by VLMs, is a promising approach (Deng et al., 2024), and our work closely contributes to this line of work. Recent approaches fine-tune VLMs (Gunjal et al., 2024) with human-annotated data, calling LLM to leverage tools like open-vocabulary detectors, OCR (Chen et al., 2024b), or estimate prediction uncertainty (Farquhar et al., 2024). Despite these methodological developments and the use of VLMs as a detector, VLMs' fundamental ability to detect hallucinations in captions is unclear due to the lack of a benchmark.

**Datasets for hallucination detection in image captioning.** Some datasets are introduced for HalDec (Wang et al., 2023b; Chen et al., 2024b; Gunjal et al., 2024; Wada et al., 2025) (Table C[2]), but they are limited as VLM benchmarks, often lacking diverse Captioners or sufficient samples per model. Our benchmark, HalDec-Bench, addresses these gaps by (i) covering more responses, (ii) balancing data across diverse models, and (iii) incorporating text-to-image outputs. SUGARCREPE (Hsieh et al., 2023) and ARO (Yuksekgonul et al., 2023) probe CLIP's fine-grained image-text alignment ability, but rely on rule-based perturbations and simple sentences. In contrast, HalDec-Bench uses VLM-generated captions, which are more challenging as shown in Sec. 4.2.

# 3 DATASETS

We aim to collect datasets that cover diverse image-caption pairs equipped with high-quality annotations of hallucination presence. This section first explains how we collect image-caption pairs and provide annotations to them, followed by an analysis of the dataset. We focus on obtaining labels for sentence-level hallucination presence for two reasons: (i) sentence-level labels give a cue to easily find more fine-detailed locations of hallucinations, and (ii) span-level annotation suffers more from the subjectivity of annotation than sentence-level. For deeper analysis, we additionally provide span-level hallucination presence labels and categorize the types of hallucinations. Due to the limited space, we leave most details in Appendix C.

## 3.1 COLLECTING IMAGE-CAPTION PAIRS

HalDec takes an image and a caption as input and decides the presence of hallucinations. Thus, coverage of diverse image domains and caption patterns is essential to building a benchmark. We thus obtain image-caption pairs using six image-to-text and two text-to-image models. This process produces image–caption pairs, where each *caption* consists of multiple consecutive sentences.

**Image-to-text models (Captioner).** We employ CC12M (Changpinyo et al., 2021) and the validation split of COCO 2017 (Lin et al., 2014) as image inputs. To ensure the diversity of the test images, we cluster images into 50 clusters and pick 40 images from each cluster, resulting in 2000 images in total. We manually categorize each cluster to enable interpretable analysis. As shown in Table A, we employ GPT-4o, ShareGPT (S-GPT) (Chen et al., 2024a), LLaVA-1.6 (Li et al., 2024a), Llama-4 (Meta.AI, 2025), Qwen 2 (Wang et al., 2024a), and CogVLM (Wang et al., 2024b), covering diverse architectures, scales, and openness. This diversity enables the collection of captions with differing levels of detail and language style. For each Captioner-image pair, we apply a chat template (e.g., "Describe the image."), yielding 12K outputs in total.

**Text-to-image models.** To ensure the diversity in a text prompt, we first pick 170 common object categories and prompt GPT-4o-mini (OpenAI, 2023) to include at least one of the categories and generate 3-4 sentences per prompt, resulting in 1000 prompts. To convert the prompts into images, we utilize Stable Diffusion 3.5 (SD) (Stability AI, 2024) as an open-source model and image generation model accessible through GPT-4o-mini (GPT-Gen), yielding 2K outputs in total.

---

[2]ZINA is concurrent with ours, and details were unavailable at submission; we compare as best we can.

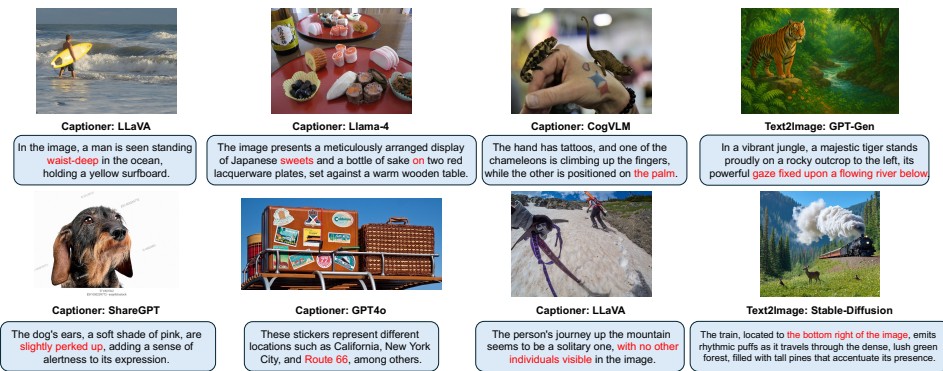

Figure 2: **Examples of hallucinated sentences in HalDec-Bench.** The hallucinated portions are often subtle, requiring fine-grained image-text alignment ability to detect them.

Table 1: **Stats of HalDec-Bench in sentence-level hallucination presence.** Our benchmark contains a large number of annotated sentences, enough for benchmarking models. We exclude sentences with *unknown* label.

| Stats | Image-to-Caption (Captioners) | | | | | | Text-to-Image | | |
|---|---|---|---|---|---|---|---|---|---|
| | CogVLM | GPT4o | ShareGPT (S-GPT) | Llama-4 | LLaVA-1.6 (LLaVA) | Qwen2 | Stable Diffusion (SD) | gpt-image-1 (GPT-Gen) | Total |
| Sentences | 7372 | 12790 | 17610 | 14800 | 15170 | 15790 | 2417 | 3524 | **89473** |
| Sentences / Image | 3.7 | 6.4 | 8.8 | 7.4 | 6.9 | 7.9 | 2.4 | 3.5 | **6.3** |
| Correct (%) | 91.5 | 91.8 | 73.4 | 85.2 | 80.9 | 88.8 | 34.1 | 79.7 | 82.7 |
| Incorrect (%) | 8.5 | 8.2 | 26.6 | 14.8 | 19.1 | 11.2 | 65.9 | 20.3 | 17.3 |

## 3.2 ANNOTATION

In the main paper, we focus on how to obtain labels for sentence-level hallucination presence and leave the fine-detailed annotation process in Appendix C. The use of SOTA models as Captioners makes the annotation non-trivial because hallucinations produced by such models are often subtle and not immediately apparent at first glance, as shown in Fig. 2.

**Annotation labels.** We are inspired by the labeling scheme of Gunjal et al. (2024), where annotators assign one of three categories: *correct*, *incorrect*, or *unknown*. A sentence is labeled *correct* if it accurately describes the image, and *incorrect* if it contains a part that does not correctly describe the image. When correctness cannot be determined—for example, if the object is too small to recognize or if the description involves non-visible attributes such as smell or wind—it is labeled *unknown*. The *unknown* category is introduced to exclude unreliable cases from evaluation.

**Annotation process.** To ensure high-quality annotations, we adopt a two-stage process for sentence-level annotation: (i) crowd-sourced workers annotate each sentence, and (ii) we review the merged outcomes to guarantee quality. In the first stage, five independent workers annotate each sentence, reducing the risk of missing hallucinations. Moreover, their performance is continuously monitored through regular checks and feedback. The results are then merged based on majority voting, as detailed in the appendix, and subsequently reviewed. During the review, to minimize the inclusion of ambiguous cases in the evaluation, sentences that are difficult to judge are labeled as *unknown*. This process creates the dataset with sentence-level hallucination presence annotations. We further annotate this dataset to provide segment-level hallucination presence labels and hallucination category labels as detailed in Appendix C, where hallucinations are categorized into eight types. These categories should reveal the weakness of the current VLMs in understanding the image content.

## 3.3 ANALYSIS OF THE DATASET

**Examples of annotated sentences.** Figure 2 presents examples of hallucinated sentences. Captioners' errors often involve visual details or object relationships rather than clear mistakes, making them harder to detect. Therefore, detectors require a fine-grained understanding of image-text alignment.

**Basic stats.** Table 1 summarizes the statistics of about 90K annotated *correct* or *incorrect* sentences. HalDec-Bench provides a balanced number of correct and incorrect sentences (excluding

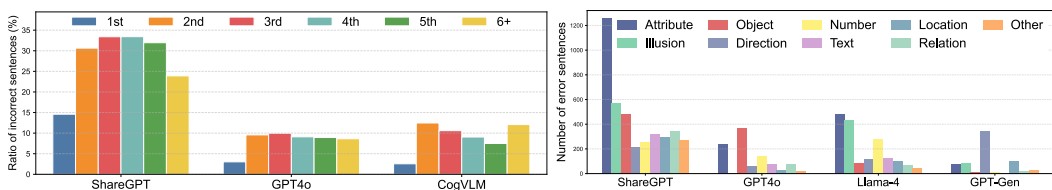

Figure 3: **Left:** Ratio of incorrect sentences within each sentence position per Captioner. Different colors indicate different positions. All models produce fewer errors at the 1st position. **Right:** Number of hallucinations for each category. Most models make many mistakes in attributes and text.

*unknown* cases). GPT-4o achieves the highest accuracy (91.8%), with only a small fraction of incorrect or unknown cases. Among open-source VLMs, Qwen (88.9%) and CogVLM (91.5%) perform comparably to GPT-4o. This suggests that for advanced Captioners, a large number of outputs are needed to obtain sufficient hallucinated samples. In contrast, models like LLaVA, LLaMA-4, and ShareGPT generate more sentences but with only 70% accuracy.

**Positions of the hallucinations.** The left of Fig. 3 computes the ratio of incorrect sentences in each position of the sentence. Across models, incorrect sentences appear most frequently in the second to fourth positions. This observation is consistent with previous work (Zhou et al., 2024; Hirota et al., 2025). The first sentence is less likely to contain hallucinations, likely because it often provides an overall image summary. In contrast, subsequent sentences typically provide finer-grained details, which are more error-prone. Beyond the sixth sentence, the error rate decreases again, as later sentences often serve as conclusions or closing remarks rather than detailed descriptions.

**Hallucination categories.** We show the hallucination type for each model on the right of Fig. 3. Errors in *attributes* are the leading category for many models, indicating that adjectival descriptions (e.g., color, texture) are prone to hallucination. Also, many models tend to cause errors in *text*, probably because small texts are hard to read even with advanced models.

## 4 EXPERIMENTS

We aim to benchmark and analyze diverse VLMs on the HalDec task to uncover key factors for building a performant HalDec model. After describing the experimental setup, we first present an overview of the empirical results, followed by a detailed analysis. In summary, we discover many notable findings: (i) HalDec-Bench offers tasks with diverse levels of difficulty, making it a strong benchmark for systematic and interpretable analysis; (ii) CLIP-like models are no longer effective to detect hallucinations generated by advanced Captioners; (iii) VLMs tend to regard sentences at the beginning of the whole caption as *correct* irrespective of their correctness; and (iv) detectors exhibit strong self-preference, consistently scoring their own outputs more favorably.

**Setups.** We aim to benchmark diverse VLMs in sentence-level hallucination detection, i.e., identifying if hallucination exists given an image and a single sentence. Specifically, each sentence and image is independently fed into VLMs. We choose this evaluation protocol since the prior work on hallucination detection (Mishra et al., 2024) also employs sentence-level evaluation. Following (Chan et al., 2023), we prompt the VLMs to output the score of the alignment between the image and an input sentence, ranging from 0 to 100, as shown in Appendix D.1. We also include BLIP-2 (Li et al., 2023a), TripletCLIP (Patel et al., 2024), and SigLIP (Zhai et al., 2023) as fundamental image-text alignment models. Given the alignment score, we compute the AUROC within each Captioner, which enables threshold-free evaluation, and the random prediction results in a score of 50.

### 4.1 OVERVIEW OF RESULTS

Table 2 presents the results evaluated on diverse VLMs. Samples of detectors' outputs are available in Fig. 8.

**HalDec-Bench covers diverse levels of hallucination detection.** We see variations in the performance across the tested VLMs and caption models. Thus, HalDec-Bench is suitable to quantify the ability of VLMs as a hallucination detector in captions.

Table 2: AUROC results across VLMs. Cells with the best performance within open-source and closed-source groups are highlighted in a blue background, while the best model within each model family is marked in bold.

| Detector | Reference | Params | Image-to-Caption Models | | | | | | Text-to-Image Models | | Avg. |
|---|---|---|---|---|---|---|---|---|---|---|---|
| | | | S-GPT | Llava | Qwen-2 | GPT4o | CogVLM | Llama-4 | SD | GPT-Gen | |
| **Open-Source Models** | | | | | | | | | | | |
| TripletCLIP | Patel et al. (2024) | 0.3B | 50.7 | 51.9 | 53.9 | 53.9 | 51.0 | 50.9 | 48.2 | 44.9 | 50.7 |
| SigLIP | Zhai et al. (2023) | 0.2B | 51.7 | 52.0 | 53.3 | 52.8 | 53.1 | 54.9 | 50.7 | 55.2 | 47.8 |
| BLIP-2 | Li et al. (2023a) | 1.2B | 53.4 | 55.9 | 52.4 | 52.5 | 52.1 | 52.2 | 48.8 | 42.1 | 51.1 |
| Phi-4 | Abouelenin et al. (2025) | 5.6B | 54.2 | 53.1 | 52.5 | 51.4 | 51.9 | 50.8 | 52.6 | 50.9 | 52.1 |
| Qwen-2 | Wang et al. (2024a) | 7B | 60.2 | 55.3 | 55.9 | 46.0 | 51.9 | 53.2 | 54.6 | 46.0 | 52.9 |
| Deepseek-VL2 | Wu et al. (2024) | 27B | 58.0 | 56.6 | 56.9 | 54.4 | 54.2 | 53.9 | 56.1 | 50.5 | 55.1 |
| LLaVA-NeXT | Li et al. (2024a) | 72B | 59.4 | 56.6 | 58.9 | 56.7 | 57.5 | 54.9 | 59.0 | 53.3 | 57.0 |
| Pixtral-12B | Agrawal et al. (2024) | 12B | 64.3 | 60.8 | 60.6 | 57.1 | 57.3 | 55.7 | 64.7 | 56.7 | 59.6 |
| Gemma-3 | Gemma Team et al. (2025) | 12B | 67.0 | 63.3 | 63.1 | 57.6 | 59.4 | 54.0 | **64.1** | **52.2** | 59.9 |
| | | 27B | **67.6** | **64.4** | **67.9** | **61.1** | **66.4** | **60.6** | 63.7 | 50.5 | **62.8** |
| InternVL2 | Chen et al. (2024d) | 2B | 55.7 | 56.0 | 58.4 | 55.7 | 60.3 | 56.1 | 52.0 | 48.5 | 55.3 |
| | | 8B | 66.6 | **65.7** | **69.1** | **63.9** | **67.6** | 60.6 | 64.3 | 52.5 | **63.8** |
| | | 26B | 63.9 | 60.6 | 61.2 | 57.1 | 58.9 | 55.6 | 53.1 | 43.8 | 56.8 |
| | | 40B | **69.3** | 63.0 | 66.5 | 59.6 | 61.9 | **61.5** | **69.5** | **56.5** | 63.4 |
| InternVL2.5 | Chen et al. (2024c) | 78B | 74.1 | 70.3 | 73.7 | 63.9 | 68.1 | 63.1 | 72.0 | 55.7 | 67.6 |
| Qwen-2.5 | Bai et al. (2025) | 7B | 68.6 | 65.3 | 66.5 | 63.4 | 64.6 | 61.0 | 65.4 | 55.7 | 62.9 |
| | | 32B | **73.6** | **71.6** | **70.6** | **66.1** | **69.0** | **66.0** | **68.9** | **61.0** | **68.4** |
| Llama-4 | Meta.AI (2025) | 109B | 80.7 | 78.6 | 77.6 | 67.5 | 77.2 | 59.9 | 81.1 | 64.7 | 73.4 |
| | | 400B | **81.1** | **80.9** | **79.0** | **71.9** | **81.3** | **64.7** | **83.0** | **67.8** | **76.2** |
| **Closed Models** | | | | | | | | | | | |
| Gemini-2.0 Flash | Gemini team (2024) | N/A | 76.8 | 73.5 | 74.9 | 65.5 | 70.4 | 64.8 | 73.4 | 57.0 | 69.5 |
| GPT4o-mini | | N/A | 69.9 | 65.7 | 68.2 | 60.0 | 62.3 | 60.3 | 56.5 | 47.7 | 61.1 |
| GPT4o | OpenAI (2023) | N/A | 75.8 | 72.6 | 72.8 | 58.2 | 69.7 | 63.8 | 63.2 | 52.4 | 66.1 |
| GPT4.1-mini | | N/A | 77.8 | 75.8 | 74.4 | 65.8 | 69.2 | 66.0 | 68.7 | 56.1 | 69.2 |
| GPT5-mini | | N/A | 81.5 | 82.2 | 80.2 | 69.7 | 81.1 | 73.0 | 83.8 | 65.7 | 77.2 |
| GPT5 | | N/A | **85.4** | **86.0** | **85.3** | **72.3** | **85.5** | **78.4** | **84.9** | **72.0** | **81.2** |

**Best model.** On average, GPT-5 shows the best performance of all models, while Llama-4, the best open-source model, performs on par with GPT-5-mini. Llama-4 outperforms many private models with a large margin. Its activated parameters during inference are only 17B. When considering the balance of inference time and accuracy, Llama-4 is the best in open-source models.

**CLIP-based models are almost blind.** TripletCLIP, SigLIP, and BLIP-2 show AUROC around 50, indicating that they cannot distinguish correct and incorrect sentences.

**Which Captioner produces hard-to-detect hallucinations?** Hallucinations from GPT-4o, GPT-Gen, and Llama-4 are difficult to detect, even for proprietary models, as shown by their low scores. Since SOTA models like GPT-4o and Llama-4 accurately understand many scenes, their errors might be subtle and harder to identify. GPT-Gen's hallucinations often involve eye direction or fine visual details, which are also challenging. In contrast, detectors achieve higher performance on ShareGPT, LLaVA, and SD, whose outputs contain many object-level hallucinations (see Fig. F).

**Increasing the model size improves performance.** In the same model family, larger language models yield better performance, probably because the task requires interpreting diverse captions.

**Robustness to text-to-image models differs by detectors.** Models such as Llama-4 and GPT-5-mini show the highest performance in SD across captioners, indicating that SD is the easiest split for these models. By contrast, for Gemma-3 (27B) and Qwen-2.5 (32B), the performance on SD is lower than S-GPT and Qwen-2. The difference should be due to the domains of images and the difference in language patterns. Inclusion of diverse data in our benchmark helps to find such trends.

## 4.2 ANALYSIS

Given the overview above, we further provide a detailed analysis of the benchmark and detectors.

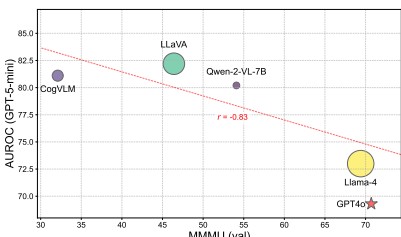 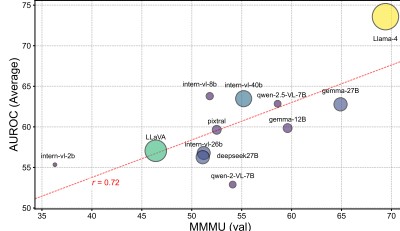

Figure 4: The size of plots indicates the parameter size. **Left:** MMMU performance measured on *Captioners* (X-axis) vs. AUROC measured by GPT-5-mini (Y-axis) for each Captioner. Advanced Captioners tend to produce hard-to-detect hallucinations. **Right:** MMMU (X-axis) vs. AUROC (Y-axis) measured on each *detector*. Detectors with better MMMU performance tend to show better performance on our benchmark.

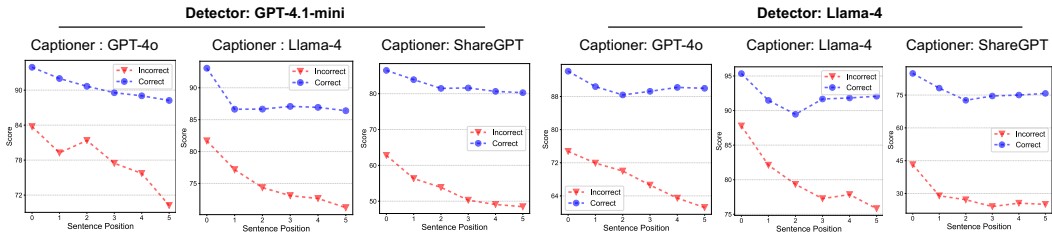

Figure 5: **Detectors show positional bias in scoring.** We average the detectors' correctness scores (Y-axis) by sentence position (X-axis) and visualize the results using GPT-4o (Left) and Llama-4 (Right) as detectors. Both detectors assign higher scores to sentences appearing near the beginning of the output. The detector is *not* provided with any positional information during inference.

Table 3: **Comparison to existing datasets** (AUROC). We compare with other HalDec datasets and the dataset used to assess VLM's compositionality understanding. Numbers of prior benchmarks that exceed HalDec-Bench are highlighted. This result indicates that HalDec-Bench is more challenging than existing datasets.

| Detector | Params | HalDec-Bench | | | | Hallucination Detection | | | VL-Compositionality | |
|---|---|---|---|---|---|---|---|---|---|---|
| | | ShareGPT | Llava | GPT4o | Llama-4 | MHalDetect | Foil | HAT | ARO | SugarCrepe |
| Qwen-2.5 | 7B | 68.6 | 65.3 | 55.7 | 61.0 | 78.7 | 85.5 | 68.0 | 78.3 | 84.9 |
| Gemma-3 | 27B | 67.6 | 64.4 | 67.9 | 60.6 | 81.7 | 91.8 | 76.6 | 79.2 | 87.6 |
| Llama-4 | 109B | 80.8 | 78.7 | 67.8 | 59.9 | 82.8 | 90.3 | 76.3 | 84.8 | 89.4 |

**Hallucinations generated by better Captioners are harder to detect.** The left of Fig. 4 plots Captioner performance on MMMU (x-axis) against AUROC measured by GPT-5-mini (y-axis). Captioners with higher MMMU scores tend to yield lower AUROC, indicating that stronger Captioners generate hallucinations that are harder to detect.

**The performance on HalDec-Bench is highly correlated with that on MMMU.** The right of Fig. 4 plots the performance on MMMU (x-axis) and HalDec-Bench (y-axis), and indicates that models effective on MMMU perform well on HalDec-Bench and vice versa.

**VLMs are biased to favor the sentence near the beginning of the output.** Figure 5 computes the detectors' output scores averaged within each sentence position. For both *correct* and *incorrect* image-text pairs, the detectors give a higher score to the sentences located near the beginning of the output. The first sentence often provides the overview of the image without details, and VLMs seem to prefer such a sentence, possibly because such sentences are abundant in training datasets.

**HalDec-Bench is more challenging than prior HalDec datasets.** Table 3 compares HalDec-Bench with prior hallucination detection and VL-compositionality datasets. We evaluate on HAT and FOIL (Petryk et al., 2024), which inject hallucinations by word replacement in human captions, and MHalDetect (Gunjal et al., 2024), which annotates outputs of InstructBLIP (Dai et al., 2023). ARO (Yuksekgonul et al., 2023) and SugarCrepe (Hsieh et al., 2023) target compositionality evalu-

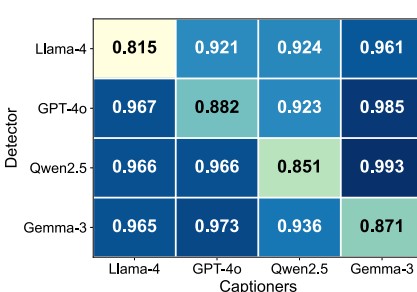 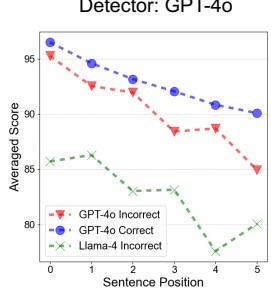

Figure 6: **Detectors struggle to detect their own hallucination. Left:** Self- and cross-evaluation results. AUROC scores for each Captioner (columns), normalized by the average AUROC of each Detector (rows). Diagonal entries show self-evaluation. **Right:** We pick GPT-4o as a detector, with their output correctness scores averaged by sentence position. Blue and red lines show scores for *correct* and *incorrect* GPT-4o's outputs; green shows scores for *incorrect* Llama-4 outputs.

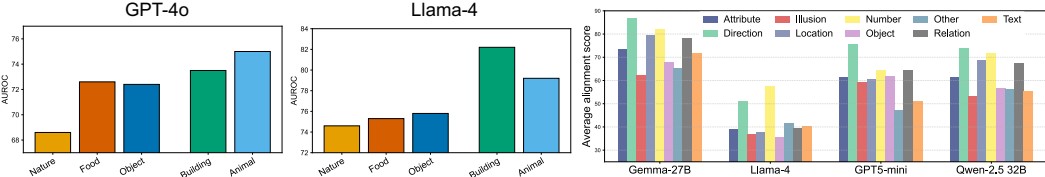

Figure 7: **Left: AUROC on different image domains.** The relative ordering of AUROC was highly consistent across models, exhibiting similar patterns of strength and weakness across domains. **Right: Detectors' score averaged within each hallucination type.** All models show weakness in *Direction* and *Number* hallucination.

ation. ShareGPT is the easiest split of HalDec-Bench, with performance close to HAT. All detectors excel on FOIL, suggesting these hallucinations are easy to detect for current SOTA VLMs.

**Detectors struggle to detect their own hallucinations.** Table 2 shows that Llama-4 (109B) and GPT-4o perform poorly on their own outputs (highlighted by underline). Their average rankings across Captioners are 3 and 6.8, respectively, but drop to 13 and 12 in detecting their own hallucinations. This finding aligns with prior work reporting LLM evaluators favor their own outputs (Panickssery et al., 2024). We annotate captions generated by Qwen-2.5 (32B) and Gemma-3 (27B) to enable more extensive self- and cross-evaluations. Figure 6 (left) confirms much lower AUROC on self-generated captions (diagonal elements). Figure 6 (right) shows that GPT-4o scores its own *incorrect* sentences higher than those of Llama-4, and the gap between incorrect and correct scores is small in its own output, which is causing the performance degradation.

**Detectors show similar domain-wise robustness.** The left of Fig. 7 studies AUROC on different image domains. The relative ordering of accuracies was consistent across models, meaning that models show similar trends in strength and weakness across domains. Performance is notably lower on *Nature*, *Food*, and *Object*. As shown in Fig. D, Captioners tend to generate accurate descriptions on such domains. Then, detecting errors from such mostly precise descriptions can get harder.

**Detectors are poor at detecting *Direction* and *Number* hallucinations.** The right of Fig. 7 assesses detectors' robustness across hallucination categories using correctness scores (lower is better since only hallucinated sentences are accounted). *Direction* errors occur when object orientation is misdescribed; identifying the errors requires fine-grained visual understanding, and detectors consistently perform poorly. *Number* errors arise from incorrect object counts—an issue long recognized in early VLMs like CLIP (Paiss et al., 2023) and still evident in advanced models. Figure 8 illustrates some results, highlighting that detectors still misunderstand the subtle visual details.

**Segment-level localization has more room for improvement.** HalDec-Bench includes hallucination segments for each hallucinated sentence, enabling segment-level evaluation. We present VLMs with a hallucinated sentence-image pair and prompt them to localize the hallucinated span, explicitly noting that one exists. Performance is measured by alignment with human annotations (see Appendix for prompts and metrics). As shown in Table 4, Llama-4 (400B), the best model, localizes

Figure 8: **Examples of *incorrect* sentences with detectors' correctness scores.** Higher scores indicate greater confidence in correctness. Detectors are prone to being overconfident in these examples. We highlight detectors' errors in red within the text and mark the grounded *incorrect* regions in the image with orange boxes.

Table 4: **Results of hallucinated segment localization task** (Average precision (%)). Localizing the segment of the hallucinated caption is challenging even for performant models.

| Detector | Params | Image-to-Caption Models | | | | | | Text-to-Image Models | | Avg. |
|---|---|---|---|---|---|---|---|---|---|---|
| | | S-GPT | Llava | Qwen-2 | GPT4o | CogVLM | Llama-4 | SD | GPT-Gen | |
| Qwen-2.5 | 32B | 17.3 | 22.4 | 14.0 | 20.8 | 22.5 | 12.6 | **15.9** | **12.1** | 17.2 |
| GPT-4o mini | - | 25.2 | 28.9 | 20.6 | 29.5 | 28.0 | 18.7 | 14.7 | 14.6 | 22.5 |
| Llama-4 | 109B | 24.9 | 27.0 | 24.8 | **34.3** | 29.3 | **19.6** | 15.1 | 11.0 | 23.3 |
| Llama-4 | 400B | **28.2** | **29.1** | **26.2** | 34.0 | **29.8** | 19.4 | 15.4 | 12.0 | **24.2** |

Table 5: **Results of model ensemble.** Ensembling detectors' outputs improves performance in almost all cases. The increase or decrease from the *better* model used for ensembling is highlighted next to each score.

| Detector 1 | Detector 2 | Image-to-Caption Models | | | | | | Text-to-Image Models | |
|---|---|---|---|---|---|---|---|---|---|
| | | S-GPT | Llava | Qwen-2 | GPT4o | CogVLM | Llama-4 | SD | GPT-Gen |
| Qwen-2.5 (7B) | Gemma-3 (27B) | 71.9 **(+3.3)** | 68.4 **(+4.0)** | 71.3 **(+3.4)** | 64.6 **(+3.5)** | 69.3 **(+2.9)** | 63.3 **(+2.3)** | 67.6 **(+2.2)** | 54.8 **(-0.9)** |
| Llama-4 (109B) | LLama-4 (400B) | 84.6 **(+3.5)** | 83.4 **(+2.4)** | 82.9 **(+3.9)** | 74.0 **(+2.1)** | 83.5 **(+2.2)** | 65.8 **(+1.1)** | 84.6 **(+1.6)** | 69.1 **(+1.2)** |
| Llama-4 (109B) | GPT5-mini | 86.0 **(+4.5)** | 84.8 **(+2.7)** | 83.6 **(+3.4)** | 73.9 **(+4.2)** | 84.4 **(+3.2)** | 72.3 **(-0.7)** | 85.2 **(+1.4)** | 68.6 **(+2.9)** |

only 24.2% of hallucinated segments on average, underscoring substantial room for improvement. Notably, GPT-4o mini outperforms Qwen-2.5 (32B), in contrast to Table 2, indicating that strong sentence-level detectors are not always effective for segment-level localization.

**Ensembling improves performance.** We examine whether ensembling improves detection. We average alignment scores from two comparably strong models (Table 2) and observe consistent gains (Table 5). This suggests that models apply distinct criteria for image-caption alignment, and combining them enhances performance. A drop occurs for ensembling Llama-4 and GPT-5-mini on Llama-4 captions, likely due to the large performance gap between the two models.

**Contents in the appendix.** Table F compares the prior approach in HalDec with VLM-based detectors, indicating that VLM-based detectors can surpass the prior one with a large margin. Table D and Table E study the effectiveness of the chain-of-thought and self-ensembling, respectively. More visualizations of annotation and detectors' output are available in Sec. F and Fig. H, respectively.

## 5 CONCLUSION

We present a benchmark, HalDec-Bench, designed to evaluate the performance of hallucination detection in image captioning. The benchmark covers diverse models and image domains, containing detailed annotations for the hallucinations. The evaluation on this benchmark reveals that HalDec-Bench contains tasks with different levels of difficulty, and is suitable for analyzing detectors. Moreover, we provide diverse interesting observations: (i) CLIP-like models are nearly blind for detecting hallucinations in this benchmark, (ii) VLMs tend to favor the sentence near the beginning of the output, and (iii) VLMs show the trend of self-preference. HalDec-Bench will become a key to establishing a more effective hallucination detector in image captions.

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

## A    LIMITATION

**Methodology.** HalDec needs to be a light-weight model, considering its application to curate datasets. However, our results indicate that VLMs with more parameters show superior performance. Also, our evaluation relies on sentence-by-sentence score output, which regards each sentence as independent. However, this protocol ignores the context of consecutive sentences. We observe that many sentences can be regarded as independent, yet considering multiple sentences together might improve the performance of hallucination detection.

**Annotations.** Judging the hallucinations in image captions involves subjective criteria of annotators. Captions may look hallucinated to some annotators, while they do not to others. Having a unified consensus on this criterion is difficult. For sentence-level annotation, we introduce a category *unknown*, which allows us to exclude such ambiguous samples during evaluation. This issue can be more significant in segment localization and categorizing hallucination types. Then, we focus on sentence-level detection to benchmark VLMs following Mishra et al. (2024).

## B    THE USE OF LARGE LANGUAGE MODELS (LLMs)

In preparing this manuscript, we made limited use of large language models (LLMs) such as Chat-GPT. Specifically, LLMs were employed only to assist with polishing the writing for grammar, clarity, and readability. No part of the research design, analysis, interpretation, or results was generated or influenced by LLMs. All scientific content, data, and conclusions are the sole work of the authors.

## C    DATASET

### C.1    IMAGE-CAPTION COLLECTION

We describe the list of models used for collection in Table A. All models except for closed ones are downloaded from Hugging Face.

Table A: Details of VLMs picked as Captioners and Text2Image models. We cover diverse models considering their size, provider, and release date.

| Model | Provider | Open/Closed | Scale | Release |
|---|---|---|---|---|
| GPT-4o | OpenAI | Closed | - | 2024/05 |
| ShareGPT (Share Captioner) | Shanghai AI Laboratory | Open | 7B | 2023/11 |
| LLaVA-1.6 (llava-next-72b-hf) | Microsoft | Open | 72B | 2024/01 |
| Llama-4-Scout (17B-16E) | Meta | Open | 109B | 2025/04 |
| Qwen2.5-VL (7B-Instruct) | Alibaba | Open | 7B | 2024/12 |
| CogVLM ( cogvlm2-llama3-chat-19B ) | Tsinghua Univ. | Open | 19B | 2024/06 |
| Stable-diffusion-3.5-medium (SD) | Stability AI | Open | 2.5B | 2024/10 |
| GPT-Gen (GPT4o-mini) | OpenAI | Closed | - | 2024/05 |

**Captioner models.** We collect data from two sources and employ two text-to-image models. The first source is CC12M, which is designed for vision-and-language pre-training and provides broad domain coverage. The second source is the COCO 2017 dataset, where we use the validation split. For both datasets, we cluster images into 50 domains based on ResNet features and then sample 40 images from each cluster, resulting in a total of 2,000 images per dataset.

For the Captioner models, we randomly select one of the following instructions:

> **Instruction given to captioner models**
>
> 1. Describe this image in detail.
> 2. Describe this image in detail. Instead of describing the imaginary content, only describe the content one can determine confidently from the image.
> 3. Provide a detailed description of the image, but only include elements that are clearly visible and verifiable.
> 4. Describe this image in detail. Minimize aesthetic descriptions as much as possible.
> 5. Provide a detailed, factual description without using emotional language.

**Text-to-image models.** We employ two text-to-image models. The first is stabilityai/stable-diffusion-3.5-medium, a diffusion-based generative model that we run locally via the Diffusers library on GPU hardware. The second is OpenAI's gpt-image-1, which is accessed through the Responses API with gpt-4o-mini acting as the controller for image generation. For both models, we use identical prompts. To encourage category diversity, we predefine 170 object categories and randomly select one to be included in each prompt. The selected category is then inserted into an instruction given to gpt-4o-mini, which produces a 3–4 sentence description following the specification below.

> **Instruction given to GPT-4o-mini for producing text-to-image prompts**
>
> I want to create prompts to generate image using text to image model. The prompts need to satisfy the following criteria.
> 1. The prompts include 3-4 sentences.
> 2. They need to describe a scene including target.
> 3. They need to describe the state of the objects, what they are doing.
> 4. They need to describe the location of the object in image, (e.g., left, right, bottom, top, etc)
> 5. They also need to describe where the objects are looking at (e.g., left, right, bottom, top, or towards some) if the object is some organism.
> Can you suggest a prompt? Please return in the form of dictionary, with a key of "prompt".
> Output:

## C.2 VOTING AND QUALITY CONTROL

We first recruited five annotators and conducted a pilot on one hundred images. The authors reviewed all annotations, and annotators who failed to meet our quality standards were not assigned further items. This process allowed us to identify trusted annotators. Each trusted annotator was then assigned between one thousand and two thousand images. The authors checked the quality for every batch of about two hundred images. If the annotations did not meet our standards, annotators were required to re-annotate before proceeding.

After the main annotation, we applied multi-round voting. Annotator-specific weights were assigned, with trusted annotators given higher weights. The aggregated votes were used to determine the final labels. For the *incorrect* (hallucination) category, we adopted a stricter rule: if one trusted annotator or two annotators labeled an item as incorrect, the authors manually reviewed it, since hallucinations are more difficult to detect reliably than correctness. Finally, the authors adjudicated all ambiguous cases. This combination of pilot screening, ongoing audits, weighted voting, and final review ensured high-quality hallucination detection annotations.

## C.3 ANNOTATOR RECRUITMENT

For the hallucination detection task, we recruited crowd annotators and offered compensation based on the phase and level of effort. On average, annotators received around $100 for completing 2,000 images during the detection phase. Since the hallucination type annotation required more careful reading and reasoning, the compensation was higher, averaging around $150 for each model output. The exact amount varied slightly depending on the annotator's country of residence. We did not

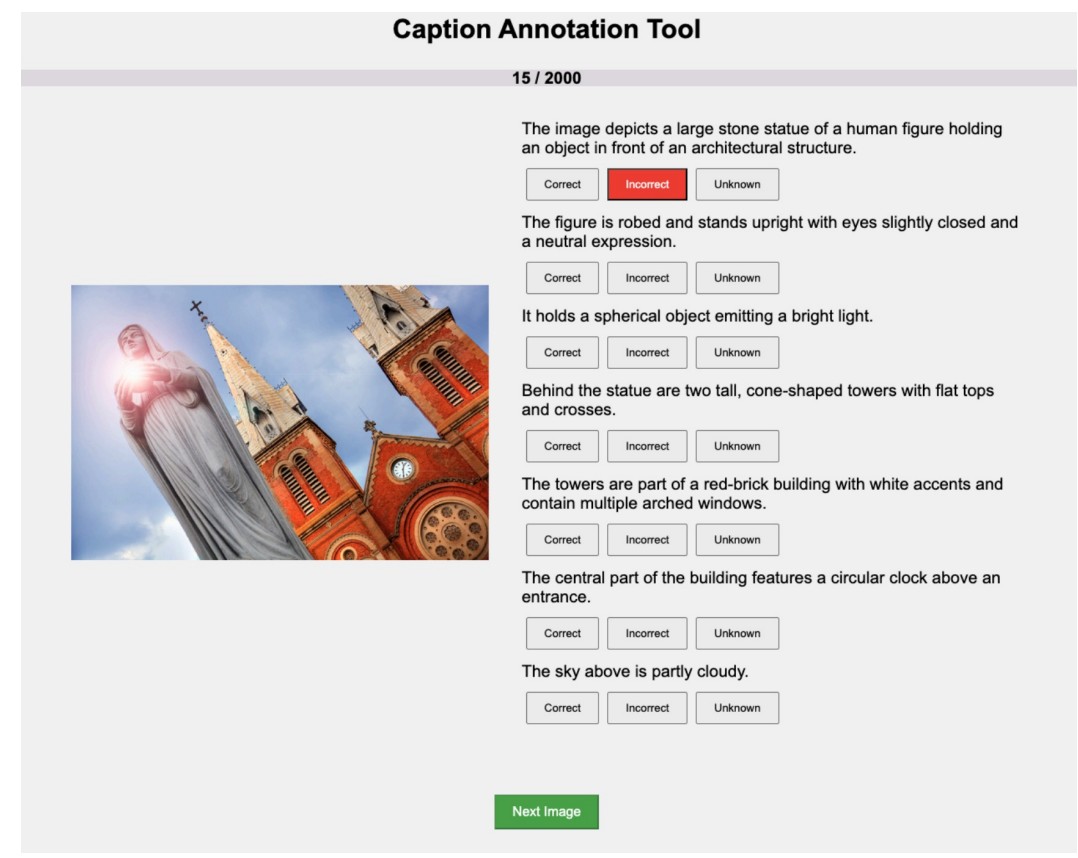

Figure A: Example of an interface used for the hallucination detection.

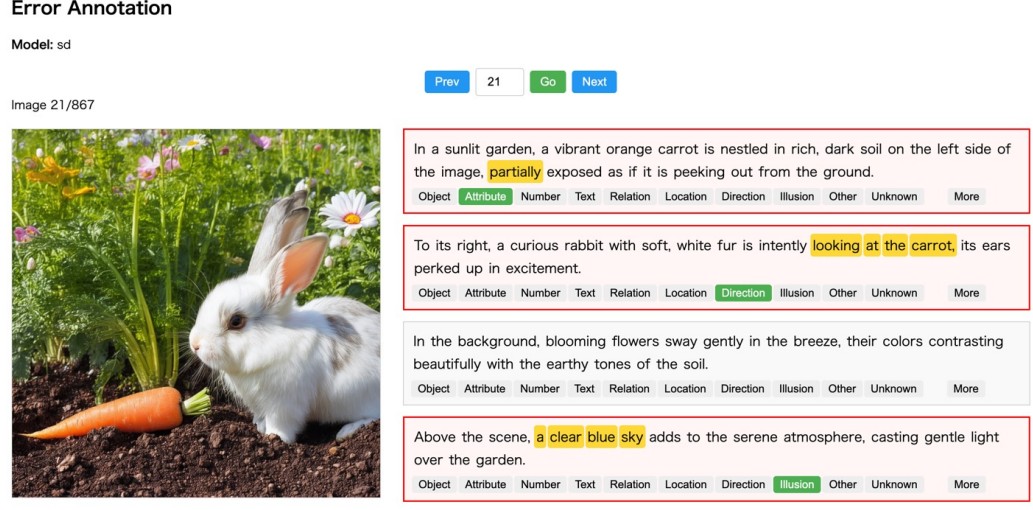

Figure B: Example of an interface used for the hallucination type annotation.

restrict annotators by location, but we required strong English reading skills, which were verified during the pilot stage. We recruited annotators on Upwork[3], Freelancer[4], and CrowdWorks[5].

---

[3] https://www.upwork.com/
[4] https://www.freelancer.com/

Table B: Types of hallucinations categorized for analysis.

| Type | Description |
|---|---|
| Object | Misidentifies an object or uses an incorrect noun (e.g., calling a dog a cat). |
| Attribute | Incorrect description of an object's property such as color, size, or action (e.g., red car described as blue). |
| Number | Incorrectly states the number of objects or people (e.g., "three people" when only two are present). |
| Text | Misreads or misrepresents textual information in the image (e.g., misreading a store sign). |
| Relation | Incorrect description of relationships between objects (e.g., "a man riding a horse" when he is standing next to it). |
| Location | Misrepresents the position of an object in the image (e.g., "a cup on the table" when it is on the floor). |
| Direction | Incorrectly describes the direction/orientation of an object (e.g., "a person facing left" when they face right). |
| Illusion | Describes objects, scenes, or actions that do not exist at all (e.g., mentioning "a flying bird" when no bird is present). |

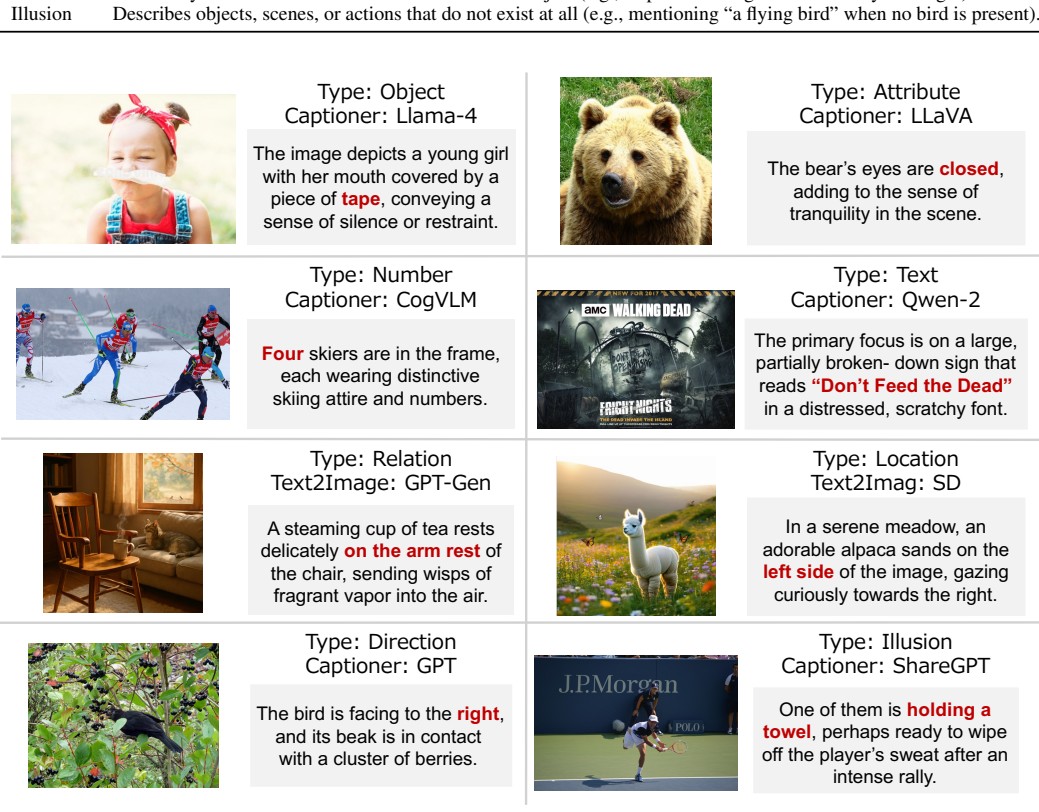

Figure C: Example annotations of error type. Hallucinations are highlighted in red.

Figure A shows the annotation interface for the hallucination detection phase, while Figure B shows the interface used for the hallucination type annotation phase.

## C.4 HALLUCINATION TYPE AND LOCATION ANNOTATION

Table B shows the eight hallucination type categories used in the HalCap dataset. These categories cover both fine-grained object- and attribute-level mistakes as well as broader contextual errors. Figure C shows annotation examples for each error type. Hallucinations are highlighted in red.

## C.5 ADDITIONAL ANALYSIS

**Detailed comparison against existing datasets.** Table C describes the detailed comparison against prior hallucination detection datasets applicable for HalDec. Our dataset includes more responses and includes text-to-image models as the evaluation target. In particular, it offers larger textual coverage, covering 1.6M words, 94k sentences, and a vocabulary of 17k unique word types, than prior datasets.

---

[5] https://crowdworks.jp/

Table C: Compared to existing hallucination detector benchmarks for image captions based on their evaluation split, HalDec-Bench offers the largest number of responses, providing annotations for at least 1,000 responses per model across eight models. This scale enables detailed, model-wise performance analysis and facilitates a deeper understanding of detector characteristics. For datasets that are not publicly available or lack information, the corresponding statistics are reported as NA.

| Dataset | Gran-ularity | # responses | # halluc. types | # models | # words | # sentences | # vocab. | # unique image | Text to Image |
|---------|--------------|-------------|-----------------|----------|---------|-------------|----------|----------------|---------------|
| HaELM (Wang et al., 2023b) | Response | 5k | ✗ | 1 | 518k | 28k | 6k | 5k | ✗ |
| MHalDetect (Gunjal et al., 2024) | Segment | 4k | ✗ | 1 | 258k | 14k | 4k | 1k | ✗ |
| MHaluBench (Chen et al., 2024b) | Segment | 1k | 4 | 5 | 15k | 1k | 2k | 1k | ✔ |
| ZINA (Wada et al., 2025) | Segment | 7k | 6 | 12 | NA | NA | NA | NA | ✗ |
| HalDec-Bench (Ours) | Segment | 14k | 9 | 8 | 1.6M | 94k | 17k | 4k | ✔ |

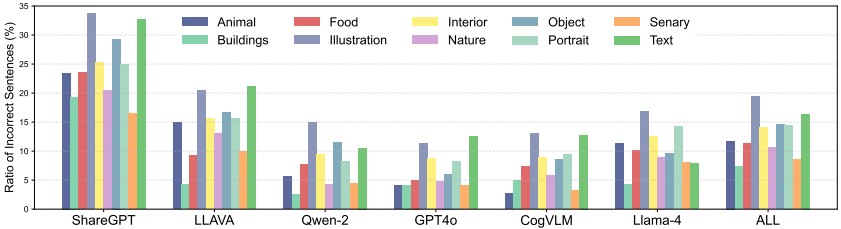

Figure D: **Ratio of incorrect sentences for each image domain.** All models tend to produce more errors in domains such as *illustration* and *Text*.

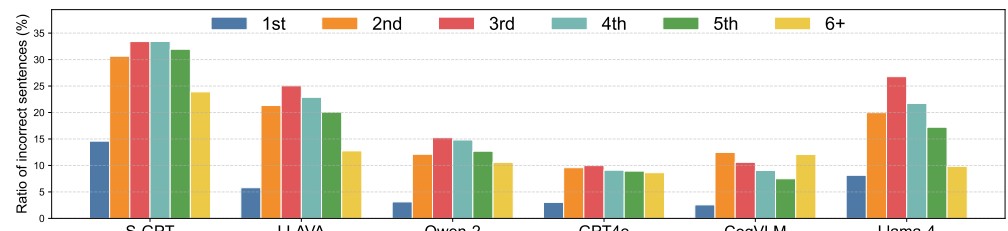

Figure E: **Ratio of incorrect sentences within each sentence position per model.** Different colors indicate different positions. All models produce fewer errors at the 1st position.

**Image domain.** Figure D illustrates the ratio of incorrect sentences on each image category in the CC12M. All Captioners tend to produce more errors in *Text* and *Illustration* domains, while they are relatively robust in real images. This can be because of the bias in the training data of the Captioners.

**Error analysis w.r.t position of the sentence.** In Fig. E, we present the ratio of incorrect sentences across sentence positions for each model. Among image captioning models, incorrect sentences tend to appear most frequently in the second to fourth positions. Interestingly, the very first sentence is less likely to contain hallucinations. This may be because the first sentence often serves as an overall image caption. In contrast, the second and subsequent sentences typically provide more detailed descriptions, which are more prone to errors. For positions beyond the sixth sentence, the error rate decreases again. These later sentences often serve as overall conclusions or closing remarks rather than detailed descriptions, which may make them similar to the first sentence and thus less prone to errors.

**Analysis w.r.t hallucination types.** Figure F describes the type of hallucinations we provide. Our dataset covers various kinds of hallucinations.

## D DETAILS OF EXPERIMENTAL SETUPS

### D.1 DETAILS OF EVALUATION

**Source of models.** We employ models available in HuggingFace and base our code on the HuggingFace Transformers package.

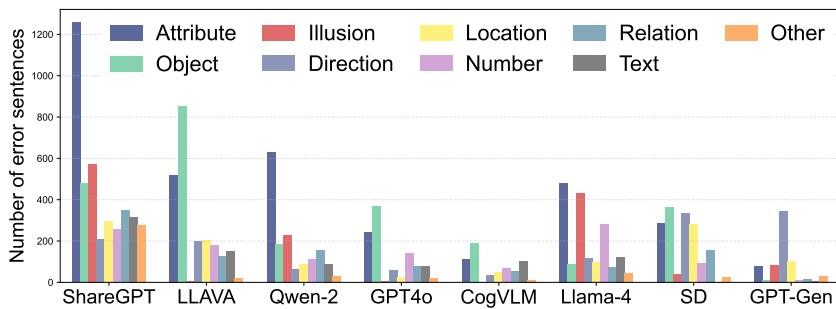

Figure F: **Number of hallucinations for each category.** Most models make many mistakes in attributes and text.

**Computation.** At most eight A100 80GB GPUs are used for inference of a single model.

**Prompt.** We employ the prompt below to compute the alignment score for decoder-based VLM.

---

**Prompt to compute image-sentence alignment**

You are given an image and a caption describing the given image. Your task is to judge if the caption describes the image correctly. If you think the sentence does not describe the image correctly, return low the score. If you think there is no mistake in the caption, return high score. Judge the correctness from 0-100 points. Return the output in the form of dictionary, e.g., "score": 50. Please first output the correctness points before explaining the reason for the score.
Caption:

---

Similarly, we use the prompt below to obtain the results of the chain of thought.

---

**Chain-of-thought prompt**

You are given an image and a caption describing the given image. Your task is to judge if the caption describes the image correctly. If you think the sentence does not describe the image correctly, return low the score. If you think there is no mistake in the caption, return high score. Judge the correctness from 0-100 points. Return the output in the form of dictionary, e.g., "score": 50. Please first explain the reason of scoring in ** two or three ** sentences and output the correctness points as shown above.
Caption:

---

**Parsing.** After obtaining the text output, we write a parser to convert the output into an integer. Models sometimes did not properly follow the prompt, and we could not parse such output. For such a sample, we assign 50 as its alignment score. In Table 2, we present models with their failure rate less than 5%. Also, the failure rate of a well-performing model is very low.

**Annotation details in self-preference analysis.** In Sec. 4.2, we additionally provide sentence-level hallucination existence labels for Qwen-2.5 (32B) and Gemma-3 (27B). To reduce the cost of annotation, we follow an annotation procedure different from the other 8 models, yet in a quality-ensured manner. Specifically, we randomly pick 500 images and generate captions using two models. Then, one quality-ensured annotator gives an annotation to 500 captions. This produces enough samples for analysis. We will include this split when publishing the dataset.

**Prompt in hallucination localization.** We employ the prompt below to obtain the results of hallucination localization.

> **Prompt for hallucination localization**
>
> You are given an image and a caption describing the given image. Your task is to localize the segment of the caption, which describes the image incorrectly. Please output the segment by marking the incorrect parts by **[]**, e.g., A **[red]** bird singing in a tree. Return the output in the form of a dictionary. Example format.
>
> ```json
> {
> "output": "A **[red]** bird singing in a tree."
> }
> ```
>
> Caption:

**Evaluation metric in hallucination localization.** We evaluate the alignment between the word spans predicted by models and the ground-truth (GT) spans using an Intersection-over-Union (IoU) based criterion. Concretely, we compute the IoU between the predicted word range and the GT word range. In Table 4, a prediction is considered correct if its IoU with a GT span is greater than or equal to 0.3. Based on this criterion, we measure precision as the proportion of predicted spans that are judged correct.

# E  ADDITIONAL EXPERIMENTS

Table D: Results of using Chain-of-Thought (COT).

| Detector | COT | Image-to-Caption Models | | | | | | Text-to-Image Models | |
|---|---|---|---|---|---|---|---|---|---|
| | | **S-GPT** | **Llava** | **Qwen-2** | **GPT4o** | **CogVLM** | **Llama-4** | **SD** | **GPT-Gen** |
| Llama-4 (109B) | | 80.7 | 78.6 | 77.6 | 67.5 | 77.2 | 59.9 | 81.1 | 64.7 |
| | ✓ | 80.6 (**–0.1**) | 80.8 (**+2.2**) | 80.0 (**+2.4**) | 71.1 (**+3.6**) | 80.1 (**+2.9**) | 62.4 (**+2.5**) | 80.8 (**–0.3**) | 65.1 (**+0.4**) |
| GPT4.1-mini | | 77.8 | 75.8 | 74.4 | 65.8 | 69.2 | 66.0 | 68.7 | 56.1 |
| | ✓ | 79.0 (**+1.2**) | 76.2 (**+0.4**) | 75.0 (**+0.6**) | 63.4 (**–2.4**) | 71.6 (**+2.4**) | 63.8 (**–2.2**) | 73.2 (**+4.5**) | 56.2 (**+0.1**) |

**Chain-of-Thought improves the performance?** Table D evaluates the impact of chain-of-thought reasoning (Wei et al., 2022), where detectors are prompted to generate a reasoning path before producing a score (see above for prompt details). For Llama-4, COT generally improves performance, whereas for some Captioners, the gains are marginal or even slightly negative. Results for GPT4.1-mini are mixed, wherein improvements highly depend on the evaluation target.

Table E: Ensembling detectors' output improves performance in almost all cases. We highlight the increase or decrease from the *better* model used for ensembling next to each score.

| Detector | Num. of Ensemble | Image-to-Caption Models | | | | | | Text-to-Image Models | |
|---|---|---|---|---|---|---|---|---|---|
| | | **S-GPT** | **Llava** | **Qwen-2** | **GPT4o** | **CogVLM** | **Llama-4** | **SD** | **GPT-Gen** |
| Llama-4-scout | 1 | 80.6 | 80.8 | 80.0 | 71.1 | 80.1 | 62.4 | 80.8 | 65.1 |
| Llama-4-scout | 5 | 83.2 (**+2.6**) | 83.0 (**+2.2**) | 81.8 (**+1.9**) | 74.4 (**+3.4**) | 82.2 (**+2.1**) | 65.1 (**+2.7**) | 83.0 (**+2.2**) | 65.9 (**+0.8**) |
| Llama-4-scout | 10 | 83.7 (**+3.1**) | 83.4 (**+2.6**) | 82.2 (**+2.3**) | 75.0 (**+3.9**) | 82.8 (**+2.7**) | 65.7 (**+3.3**) | 83.4 (**+2.6**) | 66.4 (**+1.3**) |

**Self-ensemble improves performance.** We further study the potential of ensembling. Unlike the analysis above, we ensemble outputs from a single model to refine detector's score (Farinhas et al., 2023; Jiang et al., 2023). To get different scores from a single model, we obtain different reasoning paths by stochastic sampling in the chain-of-thought. To ensure the diversity of COT, we set the temperature as 1.5 and top$_p$ as 0.9. Table E presents the results in Llama-4, where the performance consistently improves in all Captioners. Also, using more ensemble paths tends to improve the performance, while the increase seems to saturate. Model ensembling can be an interesting direction to improve the performance in this task.

Table F: Comparison to existing HalDec approaches.

| Detector | GPT4o | SD |
|---|---|---|
| UniHD (Chen et al., 2024b) | 62.6 | 71.0 |
| Qwen-2.5 32B | 66.1 | 68.9 |
| Gemma-3 27B | 61.0 | 63.7 |
| Llama-4 109B | 67.7 | 81.1 |
| GPT-4.1-mini | 65.8 | 68.7 |

Table G: Mean IoU for hallucination localization task. Localizing the segment of the hallucinated caption remains difficult even for performant models.

| Detector | Params | Image-to-Caption Models | | | | | | Text-to-Image Models | | Avg. |
|---|---|---|---|---|---|---|---|---|---|---|
| | | S-GPT | Llava | Qwen-2 | GPT4o | CogVLM | Llama-4 | SD | GPT-Gen | |
| Qwen-2.5 | 32B | 13.8 | 15.1 | 11.7 | 15.1 | 16.0 | 10.7 | 11.4 | 9.4 | 12.9 |
| GPT-4o mini | - | 21.6 | 22.4 | 18.3 | **23.3** | **21.5** | 16.4 | **12.5** | **11.9** | 18.5 |
| Llama-4 | 109B | 22.6 | 20.8 | 22.7 | 26.4 | 23.2 | 17.3 | 10.7 | 9.0 | 19.1 |
| Llama-4 | 400B | **24.8** | **22.1** | **23.3** | 26.0 | 21.7 | **18.0** | 11.9 | 9.3 | **19.6** |

**VLM detectors can surpass prior approaches.** Table F presents the comparison to UniHD (Chen et al., 2024b), which prompts LLM to utilize an open-vocabulary detector and OCR engine. The results indicate that advanced VLMs can surpass the approach without using such external tools. More detailed discussion is available in the appendix.

**Mean intersection over union in hallucination localization.** Table G shows the results of mean IoU in hallucinated segment localization. Specifically, we compute the intersection over union between the predicted and ground-truth segments and compute the average for all samples. Overall, the performance is consistent with what is reported in Table 4.

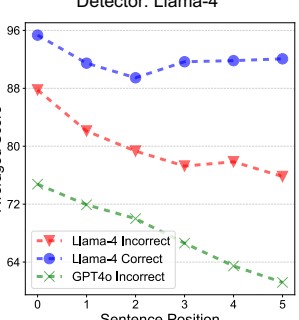
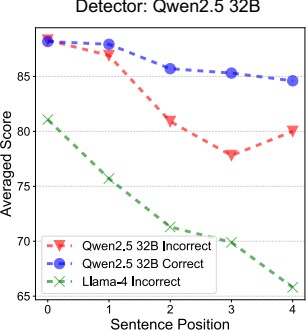

Figure G: Detector's output score for their own output captions.

**Additional results in self-preference evaluation.** Fig. G illustrates self-preference score analysis for Gemma-27B, Llama-4, and Qwen2.5. Their self-preference tendency is significant, especially for Gemma-27B and Qwen2.5.

**Additional examples of VLMs' outputs.** Figure H illustrates examples of input images, sentences, and corresponding correctness scores inferred by VLMs. VLMs tend to make errors in the location of the objects, the relationship between them, and small visual details.

# F ADDITIONAL EXAMPLES OF ANNOTATIONS

We provide additional figures illustrating annotation results and representative hallucination cases: ShareGPT (Fig. I), LLaVA (Fig. J), Qwen-2 (Fig. L), GPT-4o (Fig. L), CogVLM (Fig. M), LLaMA-4 (Fig. N), Stable Diffusion (Fig. O), and GPT-Gen (Fig. P).

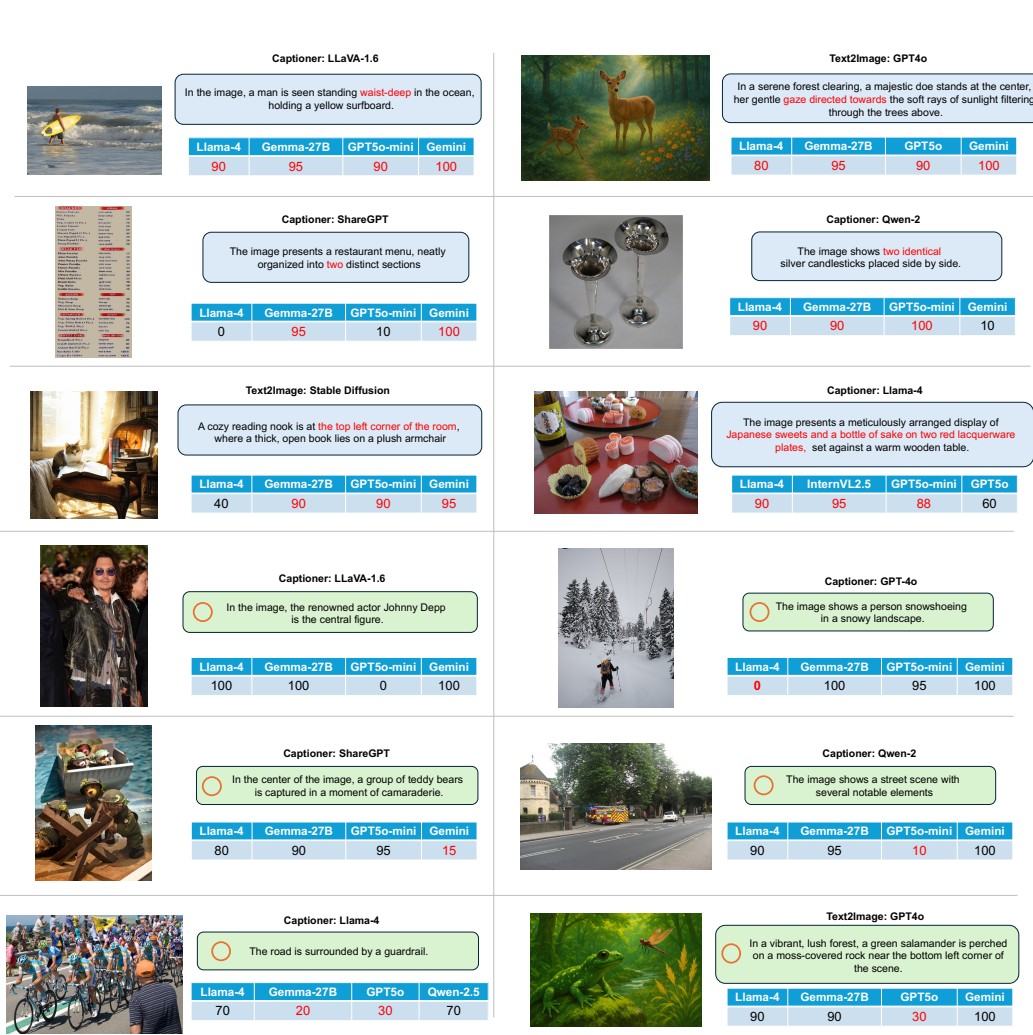

Figure H: Examples of input image and sentences with detectors' correctness scores. Higher scores indicate greater confidence in correctness. We highlight detectors' errors in red within the text.

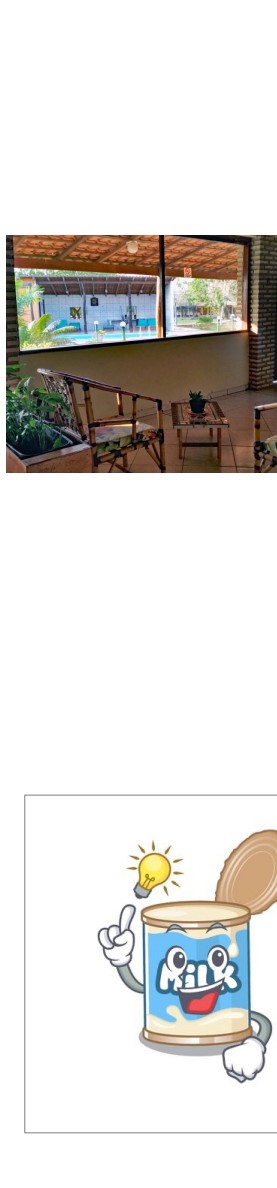

The image captures a serene and inviting outdoor patio area.

The precise placement of objects and their relative positions contribute to the overall balance and harmony of the scene.

Dominating the scene is a large window that frames a picturesque view of a pool and lush greenery, offering a glimpse into the tranquility of the surroundings.

The window is framed by a sturdy brick wall, adding a rustic charm to the setting.

**Number** In front of the window, **two** wicker chairs with vibrant floral cushions invite relaxation.

A potted plant rests on the floor next to the chairs, adding a touch of nature to the scene.

**Attribute** On the right side of the image, a painting of a bird in **flight** brings a dynamic element to the otherwise static scene.

**Attribute** The bird, captured **mid-flight**, adds a sense of movement and life to the composition.

Overall, the image presents a harmonious blend of man-made structures and natural elements, creating a space that is both functional and aesthetically pleasing.

---

In the center of a white background, there's a lively cartoon can of milk.

The entire scene is set against a stark white background, which puts the focus entirely on the can and its actions.

The image is simple yet engaging, with the can's actions and expressions adding a touch of whimsy and humor.

**Illusion** The can, painted in hues of blue and white, is anthropomorphized with a pair of arms and **legs**.

It's not just any can, though.

This one has a face, complete with eyes, a nose, and a mouth, giving it a friendly and approachable demeanor.

**Attribute** The can is actively engaged in an action—it's **holding up** a lightbulb above its head.

The lightbulb, a symbol of an idea or inspiration, is glowing brightly, indicating that the can has had a sudden thought or realization.

**Illusion** Adding to the can's character, it's wearing a **hat**.

**Illusion** The **hat** is brown, contrasting with the blue and white of the can.

**Attribute** The word "MILK" is boldly written on the can in **black** letters, standing out against the blue and white background.

---

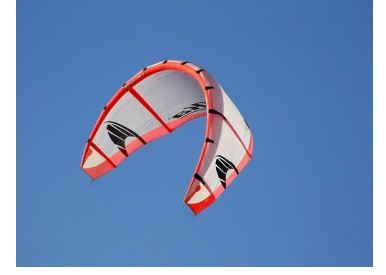
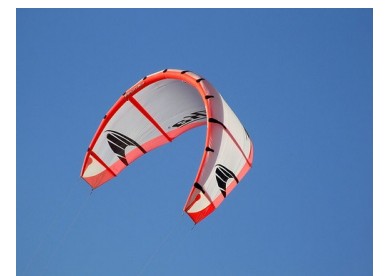

In the vast expanse of the clear blue sky, a vibrant kite dances in the wind.

The kite, a striking contrast of red and white, is adorned with black stripes that add a touch of elegance to its appearance.

**Object** It's not just any ordinary kite, but one that bears the logo of a **bird**, perhaps symbolizing freedom and soaring heights.

The kite is tethered to the earth by four strings, two on each side, held firmly by unseen hands on the ground.

These strings, like lifelines, connect the kite to its earthly roots, ensuring it doesn't stray too far from home.

**Location** The kite is positioned towards the top **right corner** of the image, as if it's eagerly reaching for the heavens.

Its position relative to the edges of the image suggests it's flying high above any potential obstructions.

This image captures a moment of joy and freedom, a snapshot of a kite's journey against the backdrop of an endless blue sky.

Figure I: Example annotations of Share-GPT.

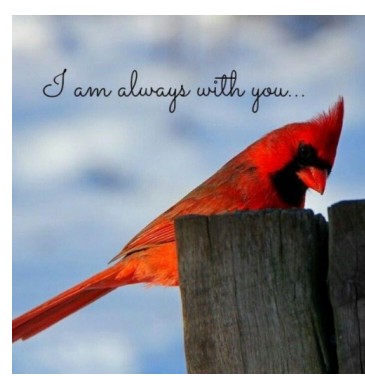

The image captures a moment of tranquility in nature.

A cardinal, with its vibrant red plumage, is perched on a wooden fence post.

**Direction**  The cardinal's head is turned **to the left**, as if it's surveying its surroundings.

**Object**  **The fence post**, weathered and gray, provides a stark contrast to the cardinal's vivid color.

The background is a serene blue sky, lightly brushed with wispy clouds.

The sky's vastness adds a sense of depth to the image.

Overlaying this peaceful scene is a quote in a cursive font that reads, "I am always with you...".

**Attribute**  The quote, written in **white**, stands out against the blue backdrop, adding a touch of sentimentality to the image.

The overall composition of the image suggests a harmonious blend of nature and human emotion.

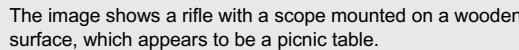

The image shows a rifle with a scope mounted on a wooden surface, which appears to be a picnic table.

The overall setting suggests a hunting or shooting context, with the equipment laid out for use or display.

The image is taken during daylight, and the focus is sharp on the rifle and the immediate surrounding items, while the background is slightly blurred.

**Direction**  The rifle is positioned horizontally, with the barrel extending **towards the left side** of the frame.

To the right of the rifle, there is a pair of binoculars with a black body and a textured grip.

**Text**  In front of the binoculars, there is a container with a label that reads **"Buckshot,"** indicating it contains shotgun pellets.

The container is open, and some of the pellets are visible.

**Relation**  **On the left side** of the rifle, there is a magazine with a label that reads "NRA," suggesting it is a publication from the National Rifle Association.

Below the magazine, there is a piece of paper with a target image and a score sheet, indicating that it might be used for target practice or competition.

The target has a bullseye and various scoring zones, and the score sheet has lines for recording the date, location, and scores.

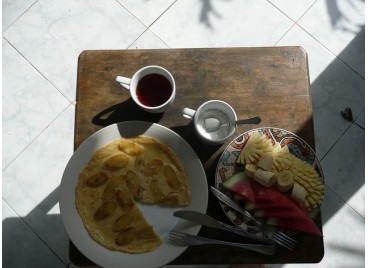

In the center of the image, a wooden table is set with a meal.

On the left side of the table, there's a white plate holding a golden brown crepe, which is garnished with slices of banana.

**Object**  The crepe is accompanied by **a small bowl of red sauce**, adding a pop of color to the plate.

On the right side of the table, there's another white plate holding a colorful fruit salad.

**Object**  The salad is a vibrant mix of red watermelon, yellow pineapple, and **green kiwi**, all arranged neatly on the plate.

In the background, there's a white tiled floor that contrasts with the wooden table.

The floor extends into the distance, creating a sense of depth in the image.

The overall scene suggests a casual and inviting dining experience.

Figure J: Example annotations of LLaVA.

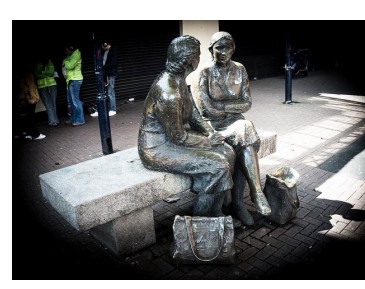

> The image depicts a bronze sculpture of two individuals engaged in a conversation.
>
> **Object** One figure, appearing to be a **man**, is seated on a stone bench, while the other, likely a woman, stands beside him.
>
> Both figures are dressed in vintage clothing, suggesting a historical or time-period-specific setting.
>
> The bench is placed on a paved area, and there are bags placed at the feet of the figures.
>
> **Number** In the background, there are **three** people wearing modern clothing, standing and appearing to be engaged in a conversation or waiting.
>
> The scene is set in a city environment, with a building and a partially open gate visible in the background.

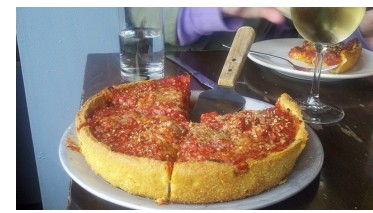

> **Attribute** The image shows a deep-dish pizza on a metal plate, with one slice **partially** removed.
>
> The pizza has a thick crust and is topped with a generous amount of tomato sauce and cheese.
>
> There is a spatula placed on the plate, likely used for serving.
>
> **Attribute** In the background, there is a person **holding** a glass of white wine.
>
> The table appears to be made of dark wood, and there is a glass of water visible next to the wine glass.
>
> The setting suggests a casual dining environment.

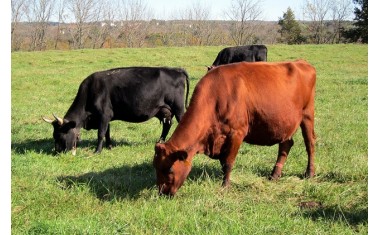

> The image depicts a serene pastoral scene of three cows grazing in a lush green field.
>
> **Attribute** The foreground prominently features a brown cow with a **white** marking on its face, which is focused on grazing.
>
> To the left, there is a black cow with distinctive white horns, also engaged in grazing.
>
> **Direction** In the background, partially obscured by the greenery, is another black cow, appearing to be standing and possibly looking in the **direction** of the camera.
>
> The field is expansive, with the cows dispersed across the landscape, suggesting a peaceful and abundant grazing environment.
>
> The background shows a mixture of trees and clear skies, adding to the natural beauty of the scene.

Figure K: Example annotations of Qwen-2.

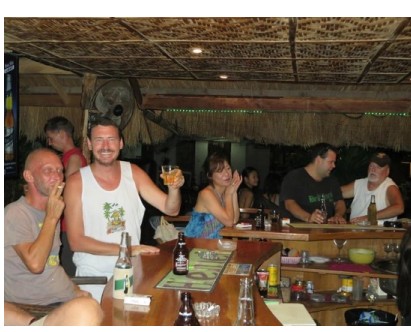

The image depicts the interior of a bar with a group of people seated around a wooden bar counter.

**Relation** The ceiling features a woven design, and a fan is mounted **above** the counter.

In the foreground, two men are sitting at the bar; one is holding a drink and appears to be smiling, while the other is holding a cigarette.

**Object** Bottles and glasses are visible on the counter, along with various bar items like **bottle openers** and condiments.

**Attribute** Further along the bar, two men and **a woman are engaged in conversation**, with one man wearing a sleeveless shirt and a hat.

The lighting is warm, giving the space a cozy ambiance.

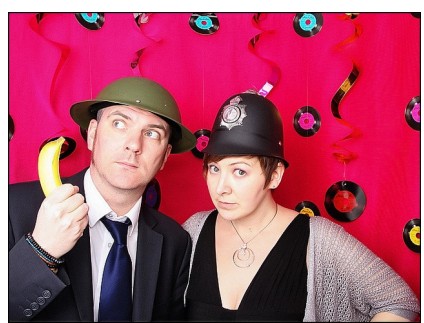

The image features two individuals posing against a bright pink background decorated with small vinyl records and hanging spiral ribbons in red and orange.

The person on the left is wearing a green military-style helmet and holding a banana.

They are dressed in a suit with a dark tie and are wearing a bracelet on their wrist.

**Object** The person on the right is wearing a black police helmet with an emblem on the front and **silver hoop earrings**.

They have short hair and are wearing a black top with a gray knitted shawl over it.

**Direction** Both individuals are looking in the **same direction** with a slightly surprised or thoughtful expression.

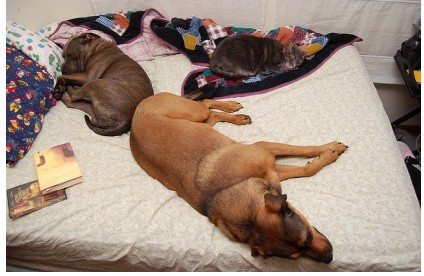

The image shows two dogs and a cat lying on a bed.

The bedspread is light-colored with a quilt at the top.

The larger dog, with a tan coat, is lying diagonally across the bed.

The smaller dog, with a darker brindle coat, is curled up near the top left of the bed.

**Relation** The cat, with a gray coat, is **nestled between them** on the quilt.

**Attribute** There are two books on the bed, one with a visible cover and one with the cover **facing down**.

The visible book cover has an image and text, but the details are unclear from the image.

In the background, there is a nightstand with various items, and curtains cover the window behind the bed.

Figure L: Example annotations of GPT4o.

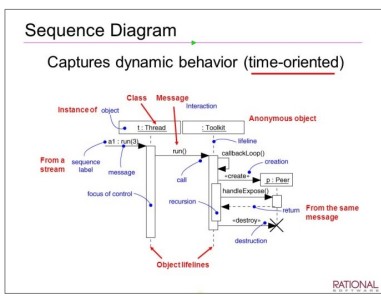

This image is a Sequence Diagram that illustrates the dynamic behavior of a time-oriented system.

**Text**

It shows interactions between different objects and classes, such as 'Instance of object', 'Class Message', 'Anonymous object', and **'Object lifetimes'**.

**Illusion**

The diagram uses various symbols like 'a1', **'a2', 'b1', 'b2'**, and arrows to indicate the flow of messages and the sequence of events.

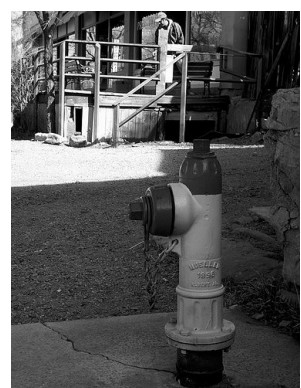

The image is a black and white photograph that captures an outdoor scene.

**Text**

In the foreground, there is a fire hydrant with the text 'MUELLER' and **'1914'** inscribed on it.

The hydrant is attached to a concrete base.

In the background, there is a wooden structure, possibly a deck or porch, with a person standing on it

**Attribute**

The person appears to be **looking out of a window or door**.

The surroundings include some trees, shrubs, and a rocky wall.

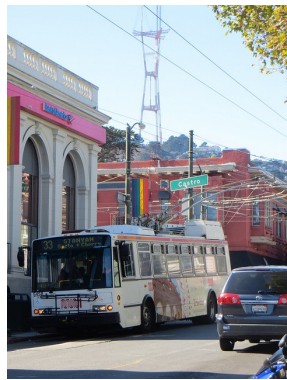

The image showcases a street scene in what appears to be an urban area.

In the foreground, there is a white bus with the number '33' displayed on its front, heading towards **'Castro'**.

The bus has an advertisement on its side.

**Text**

To the left of the bus, there is a building with a **red** facade and a sign that reads 'BankofAmerica'.

**Attribute**

The street is lined with trees, and in the background, there is a tall tower, possibly a communication or observation tower.

The sky is clear, suggesting a sunny day.

Figure M: Example annotations of CogVLM.

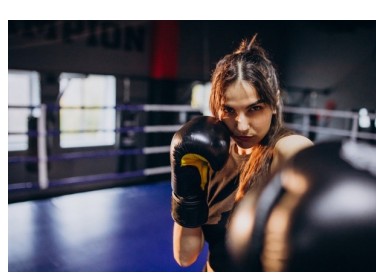

The image depicts a young woman in a boxing ring, wearing black and yellow boxing gloves.

The woman has long brown hair tied back in a pony tail and is dressed in a sleeveless top.

**Attribute** She is positioned centrally in the frame, with her **right** arm extended forward, showcasing her glove.

**Illusion** In the foreground, **another pair of boxing gloves** is visible, although they are out of focus.

The background features a blue floor and white ropes surrounding the ring, with windows on the left side that allow natural light to enter.

The overall atmosphere suggests that the woman is engaged in a training session or preparing for a fight, exuding a sense of determination and focus.

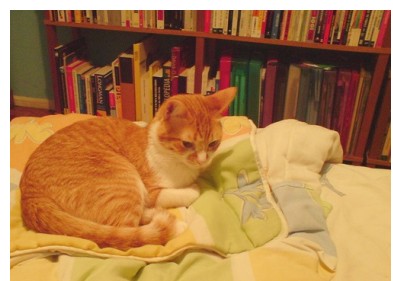

The image shows a cat lying on a bed in front of a bookshelf.

The cat is orange and white, with a white chest and paws, and an orange back and head.

**Direction** It has pointy ears and appears to be looking **at the camera**.

The cat is lying on a bed with a yellow, green, blue, and white quilt or comforter.

The bed appears to have a white sheet or comforter underneath the quilt.

In the background, there is a brown wooden book shelf filled with books of various colors and sizes.

**Attribute** The books are arranged in rows, with some standing upright and others **lying flat**.

The wall behind the bookshelf is painted light blue.

The overall atmosphere of the image suggests a cozy and comfortable setting, possibly a bedroom or study.

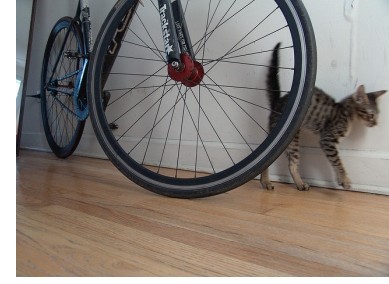

The image depicts a cat walking behind a bicycle wheel, with the cat's body partially obscured by the wheel.

The cat is positioned on a light-colored hardwood floor, facing to the right.

It has a slender build and features a brown coat with black stripes, as well as a long tail.

In the foreground, the bicycle wheel is prominently displayed, showcasing its black tire and silver rim.

**Text** The wheel is attached to a black fork with white text that reads **"Trekstar"** and other illegible words.

**Illusion** To the left of the main wheel, **another bicycle wheel** is visible, featuring a blue rim and black tire.

The overall atmosphere of the image suggests that the cat is exploring its surroundings, possibly in a home or indoor setting.

Figure N: Example annotations of Llama-4.

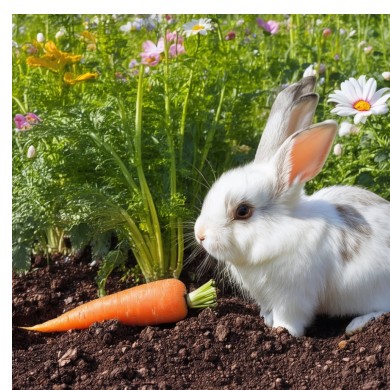

**Attribute** — In a sunlit garden, a vibrant orange carrot is nestled in rich, dark soil on the left side of the image, **partially** exposed as if it is peeking out from the ground.

**Direction** — To its right, a curious rabbit with soft, white fur is intently **looking at the carrot**, its ears perked up in excitement.

In the background, blooming flowers sway gently in the breeze, their colors contrasting beautifully with the earthy tones of the soil.

**Illusion** — Above the scene, **a clear blue sky** adds to the serene atmosphere, casting gentle light over the garden.

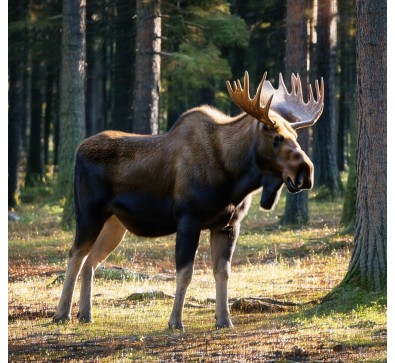

**Location** — In a serene forest clearing bathed in early morning sunlight, a majestic moose stands proudly **on the right side** of the scene, its dark coat gleaming.

**Attribute** — It **lowers its head** to nibble on the lush greenery that sprawls at its hooves while keeping a vigilant gaze towards the left, scanning for any signs of movement.

Soft rays filter through the tall pines behind it, casting gentle shadows on the dried leaves covering the forest floor.

In the background, **flickers of a sparkling stream** reflect the sun's glow as it weaves through the trees.

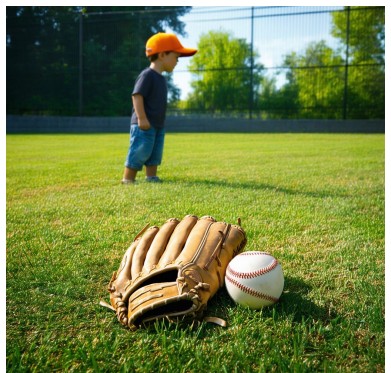

**Location** — In a sunlit park **on the left side** of the scene, an old leather baseball glove rests on the grass, slightly worn from countless games.

Next to it, a new baseball gleams in the afternoon light, ready to be thrown but currently standing motionless.

**Direction** — In the background, a young boy in a bright baseball cap stands by a fence, **looking towards the** glove with eager anticipation in his eyes, wondering when he can play catch again.

Figure O: Example annotations of Stable Diffusion.

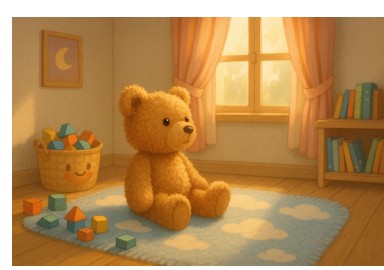

| | |
|---|---|
| | In a cozy children's bedroom, a fluffy teddy bear is nestled on the soft, cloud-patterned rug at the center of the room. |
| **Direction** | Its bright button eyes gaze thoughtfully **toward the window**, where soft rays of sunlight filter through pastel curtains, casting a warm glow around. |
| | On the left side of the scene, a pile of colorful building blocks seems to spill out of a cheerful toy basket, while on the right, a collection of books rests neatly on a shelf, hinting at adventure awaits. |
| **Attribute** | The teddy bear, **slightly tilted**, watches over the joyful mess, embodying the protective whimsy of childhood. |

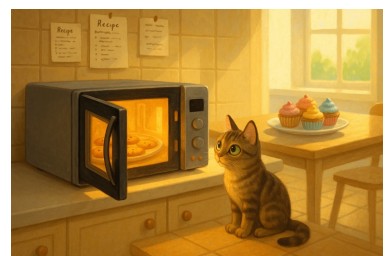

| | |
|---|---|
| **Attribute** | In a bright kitchen filled with the aroma of freshly baked cookies, a glowing microwave stands prominently on the countertop to the left, **its door slightly open** as if inviting a warm snack. |
| **Attribute** | A curious little cat with green eyes **sits on the floor** in front of it, gazing intently at the microwave's   insides, waiting eagerly for the beep that announces its treat is ready. |
| | A plate of colorful cupcakes sits on the table in the background, casting a soft shadow as sunlight filters through the window. |
| | The wall above the microwave is adorned with recipe notes, adding a cozy, lived-in feel to the scene. |

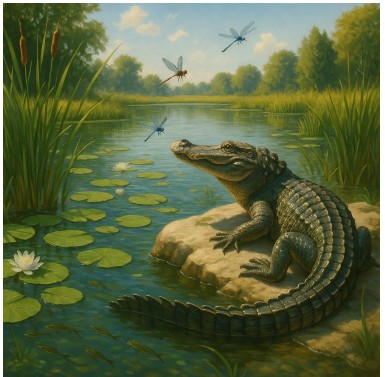

| | |
|---|---|
| | In the verdant wetlands of a sultry summer's afternoon, a crocodile lounge on a sundrenched, flattened rock at the right-hand side of the scene. |
| | Its muscular body is soaked and dripping with water, remaining vigilant as it scans the shimmering pond that stretches outward n front of it. |
| **Direction** | Surrounded by reeds and lily pads, its eyes glisten in the sunlight as it looks **towards tiny fish** darting happily beneath the surface, captivated by movement right below. |
| **Illusion** | Meanwhile, colorful dragonflies flit hip high in the air, **casting fleeting shadows** on this eager predator's competent posture. |

Figure P: Example annotations of GPT-Gen.

