# OpenReview forum: "HalCap-Bench: Benchmarking Hallucination Detector in Image Captioning"
_ICLR.cc/2026/Conference — ICLR 2026 Conference Withdrawn Submission_

### Official Review · Reviewer_JAZP · 2025-10-25

**Soundness:** 3
**Presentation:** 3
**Contribution:** 2
**Rating:** 4
**Confidence:** 5

**Summary:**

This paper introduces a new benchmark dataset for assessing image caption hallucination detectors. The authors construct a large set of image-caption pairs by using diverse models, including image-to-text models and text-to-image models. Then, human annotators review the sentence-wise alignment between images and captions. Through experiments, the authors provide the following three observations as their main contributions: 1) CLIP-based models are ineffective at detecting hallucinations within image captions; 2) VLMs tend to regard sentences near the start of an image caption as factual; 3) VLMs strongly prefer self-generated image captions.

**Strengths:**

1. This paper is clear and well-organized overall.
2. I'd like to thank the authors for their labor-intensive, human-involved benchmark construction.

**Weaknesses:**

1. Existing works have already demonstrated that CLIP shows limitations in testing alignment between images and captions [1,2]
2. According to several recent studies [3,4], image captions generated by VLMs are likely to contain factual information in their former part, and the possibility of hallucination goes up as models generate longer. Therefore, Figure 5 can be regarded as a result in support of the previous finding, not the one caused by the bias of VLMs.

[1] Yuksekgonul et al., "When and why vision-language models behave like bags-of-words, and what to do about it?" ICLR 2023
[2] Lin et al., "Evaluating Text-to-Visual Generation with Image-to-Text Generation" ECCV 2024
[3] Zhou et al., "Analyzing and mitigating object hallucination in large vision-language models" ICLR 2024
[4] Lee et al., "Toward Robust Hyper-Detailed Image Captioning: A Multiagent Approach and Dual Evaluation Metrics for Factuality and Coverage" ICML 2025

**Questions:**

Could you provide a summary of this paper’s findings that are not yet known to the community?

---

### Official Review · Reviewer_7UvM · 2025-10-31

**Soundness:** 3
**Presentation:** 3
**Contribution:** 3
**Rating:** 4
**Confidence:** 3

**Summary:**

The paper proposes a benchmark for detecting hallucinations in image captioning. The benchmark is built from captions generated by different vision–language models and includes human annotation efforts to label the correctness of each sentence in the captions with respect to the corresponding images.

**Strengths:**

- A comprehensive study that includes both **natural image–text pairs** and **AI-generated image–text pairs**.
- Incorporation of **crowdsourced human annotations**, adding reliability to the benchmark.

**Weaknesses:**

- Limited analysis on text-to-image samples:  Although the benchmark includes samples from text-to-image models, the analysis is limited. It would strengthen the work to categorize hallucination types within these generations and discuss how they differ from natural image–text cases.

- Undefined task difficulty:  The paper claims to design tasks of varying difficulties but does not define or quantify difficulty levels, which weakens the interpretability of results.

- Human agreement analysis missing:  Since the crowdsourcing annotation relies on majority voting from five annotators, it would be useful to analyze model performance under varying levels of human agreement (e.g., strong vs. weak consensus).


- Inconsistent results:  Some reported results are questionable. For instance, in Table 2 (line 276), the average AUROC for SigLIP is reported below 50, while all individual AUROC values across models exceed 50. This inconsistency should be clarified.


- Lack of visualization for correctness score distributions:  The paper could visualize the correctness score distributions (for a single detector across different caption sources) to better understand potential distribution shifts. Besides, the scores are evaluated only for sentences from one captioner. Reporting AUROC results across correct/incorrect samples from different models would clarify this effect.


- Limited model diversity:  The closed source image-to-caption models currently include only GPT-4o. It would be beneficial to include more closed-source models such as Gemini 2.0 Flash, which has been used as a detector but not as a generator.

**Questions:**

Please see **Weaknesses**

---

### Official Review · Reviewer_sgiP · 2025-11-01

**Soundness:** 4
**Presentation:** 3
**Contribution:** 3
**Rating:** 8
**Confidence:** 3

**Summary:**

The authors present a major data set for hallucination detection for (present) state of the vision language models. The set up is that 2k images are collected from CC12M and COCO val and captioned by in total six captioning models of diverse sises and strength. The resulting 12k captions are augmented with 2k generated images that stem from image generators operating on pre-produced captions. The images are grouped in several topical clusters. The obtained data sets of image - caption pairs are then annotated for containing hallucinations. Labeling is done by five fold crowd sourced human annotation plus review, guaranteeing high quality labels. Also 9 different annotation types are assigned. Hallucination is measured on sentence level.
The performance 13 open source and 6 closed source models is measured on the HalCap benchmark and subsequently analyzed. The models are prompted to score the image-text alignment. The AUROC from this score serves as metric. Among other analyses, the detection performance depends on  position of the sentence,  whether it is an detection effort from the same model as the one providing the captions.
The benchmark is compared to previous data sets for hallucination detection and stands out in size and difficulty.

**Strengths:**

- This is a really thoroughly composed data set that exceeds prior benchmarks in size and level of difficulty
- The human, crowd sourced labeling of correct vs incorrect captions is carried through with considerable quality checks
- The various annotations enable numerous detail studies
- The paper is very well prepared and graphics are instuctive
- An extensive appendix provides numerous more details

**Weaknesses:**

- The design of HalDec works with the present state of the art captioners, but as time will move on, detecting these hallucinations will be less relevant - so I expect the results to age quickly.
- The analysis of the position of the sentence is instructive, but here one would expect an investigation of the effect of reasoning. The effect might be due to short contexts at the start of the decoding from the detector.
- I might have overlooked this, but is there an analysis of the performance on the synthetically generated images vs the natural images? If not, I find his to be missing.
- The HalDec benchmark does not provide reproducible internal states of the captioning models and therefore does not enable self-monitoring methods for hallucination detection during decoding

**Questions:**

- is it planned to provide a project page with leaderboard to support the community?
- Could you detail a little more the clustering procedure
- What is the meaning of the last sentence in the caption of figure 5?
- What is the conceptual difference of the right panel in figure 6 to figure 5?

---

### Official Review · Reviewer_QSAu · 2025-11-04

**Soundness:** 3
**Presentation:** 3
**Contribution:** 3
**Rating:** 6
**Confidence:** 3

**Summary:**

The paper introduces HalDec-Bench, a benchmark to evaluate hallucination detection (HalDec) for image captions across diverse captioners, image domains, and hallucination types. Using a uniform scoring protocol where VLMs rate image–sentence alignment from 0–100, the authors report varied difficulty across splits and several consistent phenomena: (i) CLIP-style alignment models hover near chance, (ii) detectors over-trust early sentences in a caption (positional bias), and (iii) detectors show self-preference.

**Strengths:**

- The paper focuses specifically on hallucination detection in captions (not general QA), filling a gap and enabling apples-to-apples comparisons.
- This work includes multiple captioners, image domains, and sentence-level labels with hallucination types—useful for fine-grained analysis.
- The result identifies positional bias and self-preference across models. It also shows that CLIP-like models are near-random, which is actionable for practitioners.

**Weaknesses:**

- Limited context modeling. Evaluation scores sentences independently, discarding inter-sentence context that may help detect subtle inconsistencies.
- Despite multi-stage quality control, deciding hallucinations, especially span/type, can be subjective and is acknowledged as a limitation.

**Questions:**

None.

---

### Note · Authors · 2025-11-12

I have read and agree with the venue's withdrawal policy on behalf of myself and my co-authors.